# Chinese influence in Africa by integrated regret theory and multi-criteria decision analysis

**Chia-Nan Wang**[1], **Nhat-Luong Nhieu**[2*], **Ching-Ju Lu**[1]

**1** Department of Industrial Engineering and Management, National Kaohsiung University of Science and Technology, Kaohsiung, Taiwan, **2** College of Technology and Design, University of Economics Ho Chi Minh City, Ho Chi Minh City, Vietnam

\* luongnn@ueh.edu.vn

## Abstract

This study illuminates the multifaceted influence of Chinese in Africa, driven by the imperative to understand the strategic and economic ramifications of this rapidly evolving relationship. Motivated by the critical role Africa plays in global geopolitics and resource dynamics, alongside Chinese expanding international influences, the research aims to quantitatively and psychologically assess the decision-making processes underpinning this engagement. Adopting a regret theory-based Multiple Criteria Decision Making (MCDM) framework, the study evaluates Chinese impact across 49 African countries from 2018 to 2022, employing six economic indicators to capture the breadth of Chinese activities. Through meticulous normalization, regret utility computation, and total gap analysis, the methodology affords a systematic ranking that reflects the varying degrees of Chinese economic influence. The findings uncover pronounced variances in the level of Chinese engagement across the continent, with countries like Nigeria and Egypt showcasing substantial influence convergence with the theoretical model of ideal economic partnership, whereas others like Cape Verde indicate minimal influence. Contributing to academic and practical discourse, this study not only provides a methodological blueprint for analyzing geopolitical influences but also offers insights that policymakers can leverage to optimize their engagement strategies with Chinese. It foregrounds the interplay between empirical economic data and behavioral economics within international relations research. The study acknowledges limitations, primarily in data availability, which may not capture the full scope of informal economic interactions. It proposes further research to enrich the understanding of the Chinese-Africa nexus through longitudinal studies, integration of qualitative data, and expansion of the analytical model to encompass broader socio-economic impacts and more diverse indicators.

**Data availability statement:** All relevant data are within the manuscript and its Supporting Information files.

**Funding:** This research is partial funded by University of Economics Ho Chi Minh City, Vietnam. The funders had no role in study design, data collection and analysis, decision to publish, or preparation of the manuscript.

**Competing interests:** The authors have declared that no competing interests exist.

## Introduction

### Motivation

Africa holds a significant place on the global stage due to its vast natural resources, strategic geographic position, and rapidly expanding demographic profile. The continent is home to over 30% of the world's reserves of key minerals such as platinum, diamonds, and gold, along with substantial oil and gas resources [1]. These factors position Africa as a major player in the global mining and energy sectors. Geographically, Africa's coastlines along the Atlantic and Indian Oceans place it at the center of global maritime trade routes, further enhancing its strategic importance [2,3]. Demographically, Africa's youthful population—expected to double by 2050—offers immense potential as a future labor force and consumer market [4]. This shift presents new opportunities for economic transformation and increased geopolitical relevance. Additionally, the continent's biodiversity and cultural diversity contribute to ecological sustainability and global cultural heritage [5].

Chinese influence in Africa has grown considerably in recent decades, reflecting its strategic ambition and global outreach [6]. China is now one of Africa's largest trade partners and investors, with major infrastructure projects (roads, railways, ports) forming a central component of its economic diplomacy [7,8]. As illustrated in Fig 1 and Fig 2, China's footprint in sectors such as mining, telecommunications, and energy continues to reshape the economic architecture of numerous African nations [9]. China's approach is commonly framed as emphasizing economic cooperation and infrastructure investment without overt conditionality related to domestic political reforms—unlike traditional Western development models that often tie aid and loans to governance or policy change benchmarks [10]. However, this approach also raises concerns about dependency, sovereignty, and long-term development trajectories [11].

### Defining influence

In the context of international development and strategic engagement, "influence" is a multidimensional construct that encompasses a nation's ability to shape the behavior,

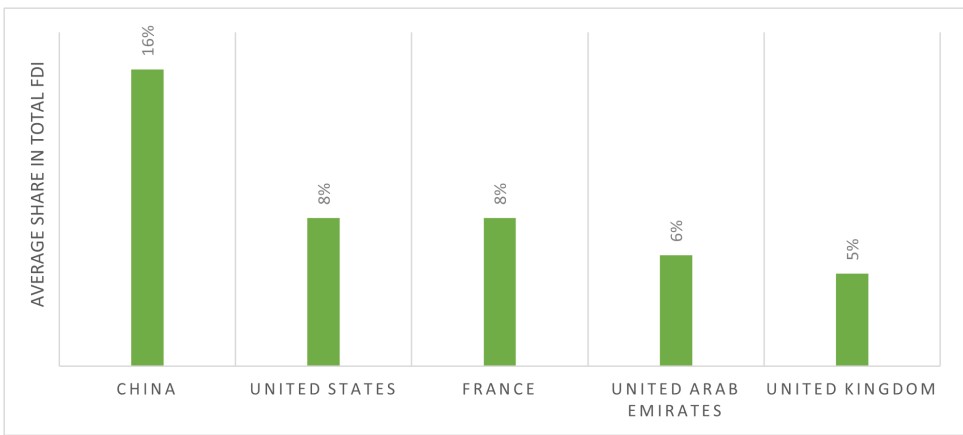

**Fig 1. Foreign direct investment (FDI) into Africa between 2014 and 2018 [7].**

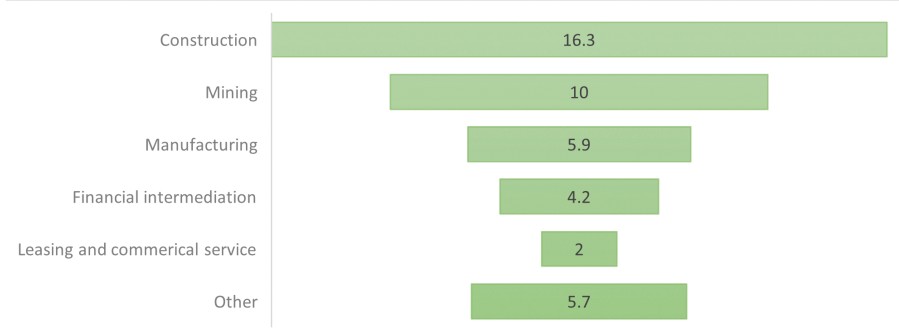

**Fig 2. The FDI stock from Chinese into African countries in 2021 [9].**

decisions, or outcomes in another country through sustained economic, political, and institutional engagement [12,13]. In this study, China's influence in Africa is not viewed as a singular, coercive force but rather as a layered phenomenon comprising both tangible and intangible elements. These include capital investments, infrastructure projects, trade relationships, labor mobility, and concessional finance. Influence is thus understood not merely in terms of power projection, but in terms of economic embeddedness, institutional dependency, and strategic alignment over time.

Such influence is often asymmetrical, shaped by the disparity in economic size, technological capabilities, and access to global markets [14]. African nations may exhibit varying degrees of responsiveness to Chinese engagement depending on their domestic priorities, governance capacities, and long-term development strategies. As a result, the impact of Chinese activities is not uniform across the continent [15,16]. For some countries, China's role may reinforce national development goals, while for others it may lead to vulnerabilities or increased dependency. Therefore, this study approaches influence as a relative and dynamic phenomenon, evaluated through multiple indicators that reflect both direct and structural forms of engagement. The goal is to empirically assess this variation and offer a more granular understanding of how African countries interact with and respond to China's multidimensional presence.

## Theoretical foundation

To capture the complexity of China-Africa relations and account for both empirical and behavioral dimensions of influence, this study adopts an integrated decision-making framework that combines four complementary components: Multiple Criteria Decision Making (MCDM), Logarithmic Percentage Change-driven Objective Weighting (LOPCOW), Multi-Attributive Ideal-Real Comparative Analysis (MAIRCA), and Regret Theory. Each of these components plays a distinct role in the analysis, collectively enabling a more comprehensive evaluation of strategic influence that reflects real-world complexity and bounded rational behavior.

MCDM serves as the overarching analytical structure, supporting evaluation in contexts where multiple, and often conflicting, criteria must be balanced simultaneously [17]. Within this framework, LOPCOW is employed to calculate objective weights for each criterion by considering their variability and impact through logarithmic percentage changes [18]. This ensures that the importance of each indicator reflects its real-world sensitivity and contribution to differentiation across countries. MAIRCA complements this by facilitating the comparison of observed country performance against an ideal benchmark, producing a ranking of alternatives that reflects their alignment with desirable strategic outcomes [19,20]. Regret Theory is introduced to simulate anticipatory behavioral responses [21,22], acknowledging that state actors—through their institutions—may seek to minimize future regret when navigating trade-offs in geopolitical partnerships. Rather than assuming perfect rationality, the framework reflects boundedly rational decision-making under uncertainty and constraint [23,24]. The integration of these methods allows for a nuanced and behaviorally informed assessment of how

African states weigh the benefits and risks of deepening their engagement with China. A summary of each method's role is provided in Table 1 below.

## Justification of indicators

To empirically evaluate the influence of China across the African continent, six key indicators were selected, each capturing a distinct dimension of engagement. These include: Chinese foreign direct investment (FDI) stock, reflecting long-term capital presence and strategic stakeholding in national economies [25]; value of Chinese exports to Africa, representing the extent of trade dependence on Chinese goods and services [26]; value of African exports to China, which measures reciprocal trade flows and resource-driven linkages [27]; number of Chinese laborers, used as a proxy for China's operational deployment and labor-export strategy [28,29]; gross revenues from Chinese construction projects, signaling the depth of infrastructure integration [30]; and loan amounts from China, indicative of financial leverage and debt diplomacy [31]. Each of these indicators is directly linked to how embedded, dependent, or strategically aligned a country becomes through engagement with China.

These criteria were selected not only based on data availability across the 2018–2022 period but also for their conceptual alignment with the study's definition of influence. Together, they allow for a holistic view that encompasses financial flows, trade asymmetries, physical presence, and long-term institutional exposure. In doing so, the framework avoids relying solely on trade or investment as a proxy for influence and instead captures a more complete picture of bilateral engagement. While prior studies have commonly measured China's influence through trade volumes [26] or foreign direct investment [32], such approaches have been critiqued for their limited scope. Scholars have noted that these proxies fail to account for the broader channels of engagement—such as labor deployment, infrastructure dominance, and loan-based conditionality—that define strategic influence in many African contexts [33,34]. By applying objective weighting and comparative ranking across these indicators, the study is able to quantify strategic alignment in a way that is replicable and analytically transparent. This multidimensional structure ensures that influence is assessed through a balanced and theoretically grounded set of metrics.

## Research objective

The overarching objective of this study is to develop and apply a novel methodological framework to systematically evaluate Chinese influence in Africa through a combined behavioral and quantitative lens. Specifically, this research seeks to integrate LOPCOW, MAIRCA, and Regret Theory within a multi-criteria decision-making (MCDM) structure to analyze 49 African countries over the period from 2018 to 2022. This integration is designed to model both the objective dimensions of influence—through investment, trade, labor, construction, and financial engagement—as well as the subjective behavioral processes involved in decision-making under uncertainty, such as the anticipation of regret.

By applying LOPCOW, the study assigns dynamic and data-driven weights to each influence criterion, while MAIRCA is used to compare real-world outcomes to ideal engagement scenarios, resulting in a ranked assessment of countries based on their alignment with Chinese influence. Regret theory further extends this by capturing how decision-makers

**Table 1. Summary of method integration.**

| Method | Role in Framework | Output |
|---|---|---|
| LOPCOW | Calculates objective weights based on criterion dispersion | Weight vector for criteria |
| MAIRCA | Ranks alternatives by comparing real vs. ideal solutions | Country-level performance scores |
| Regret Theory | Models bounded-rational preferences and simulates regret-based behavior | Regret-adjusted gap values |

might behave when faced with uncertain outcomes, trade-offs, and opportunity costs—simulating the bounded rationality that often characterizes geopolitical decision-making. The study's aim is not only to produce rankings but to offer deeper insights into the structure, pattern, and consequences of Chinese involvement. Ultimately, the research intends to generate actionable recommendations for African policymakers to maximize benefits, reduce strategic risks, and better understand the behavioral dimensions of foreign engagement, while also contributing to theoretical advancements in international political economy, behavioral decision science, and development strategy.

## Literature review

### Chinese-Africa relations studies

The literature on China–Africa relations present a diverse range of analyses addressing the economic, environmental, diplomatic, and socio-political dimensions of China's expanding presence across the continent. Numerous studies have explored the implications of Chinese trade, investment, infrastructure projects, and soft power strategies, with attention to both macroeconomic impacts and local-level transformations. Table 2 summarizes key contributions to this growing body of work.

Collectively, these studies demonstrate that China's influence is not monolithic but context-dependent, with outcomes varying based on factors such as governance quality, institutional capacity, and sectoral focus. While some research highlights positive developmental gains through infrastructure and trade (e.g., poverty reduction, increased connectivity), others raise concerns about environmental degradation, debt dependence, and asymmetrical power dynamics. Recent studies have also begun to investigate African agency, highlighting how domestic actors shape the outcomes of Chinese engagement, and how cultural and educational diplomacy is being used to cultivate long-term strategic relationships.

**Table 2. The Chinese-Africa relations studies.**

| Author(s) | Year | Research Problem | Methodology |
|---|---|---|---|
| Tawiah, V.K. et al. [35] | 2021 | Impact of different routes of the China-Africa relationship on the environment. | Fully Modified Ordinary Least Square (FMOLS) model |
| Miao, M. et al. [32] | 2020 | Impacts of China-Africa trade and Chinese FDI on the growth of African countries. | Two-step system Generalized Method of Moments (GMM) |
| Kenneth Kalu [36] | 2021 | Examination of China's public diplomacy in Africa through social and political philosophy. | Qualitative analysis based on the theory of respect and recognition |
| Amoah, P.A. [37] | 2021 | Micro-level implications of China-Africa relations on populations and individuals. | Analysis of primary and secondary data sources |
| Lau, R.K.S. [38] | 2020 | Role of African countries in China's Belt and Road Initiative. | Qualitative analysis |
| Dubinsky, I. [39] | 2021 | Examination of China's stadium diplomacy in Africa. | Qualitative analysis of media sources |
| Calabrese, L. & Tang, X. [40–42] | 2023 | Whether Chinese firms contribute to or hinder economic transformation in Africa. | Scoping review of over one hundred sources |
| Chiyemura,F. et al. [41] | 2023 | Role of African actors in shaping infrastructure projects with Chinese participation. | Context-sensitive analysis of various spheres of state agency |
| Zhang, L. et al. [42] | 2023 | Effectiveness of Chinese infrastructure investments in reducing poverty in sub-Saharan Africa. | Spatiotemporal estimation strategy using geo-coded data and demographic surveys |
| Chen, X. et al. [33] | 2024 | Impact of China's foreign direct investment and trade on green growth in Africa. | Fixed effects model, two-stage least square, and system generalized method of movement |
| Tsikudo, K. [43] | 2024 | Shift in China-Africa cooperation towards educational and cultural investments. | Analysis based on the relational productive power framework of knowledge production |

Despite this growing literature, a significant research gap remains in understanding how African nations make strategic decisions regarding their engagement with China—particularly under conditions of complexity, uncertainty, and conflicting national interests. Most existing works emphasize economic performance or qualitative political narratives, while few employ decision-analytic frameworks capable of capturing multi-criteria trade-offs or modeling bounded rational behavior. Additionally, the psychological dimension of decision-making—such as the role of anticipated regret or strategic hesitation—has received limited attention. Addressing this gap, the present study introduces a novel behavioral-MCDM approach that integrates LOPCOW, MAIRCA, and Regret Theory to assess how African countries navigate the trade-offs and uncertainties inherent in their relationship with China.

## The MCDM and bounded rationality

In complex decision environments such as international cooperation and geopolitical strategy, decision-makers must often balance multiple, conflicting criteria under conditions of uncertainty. MCDM provides a structured approach for navigating such complexity by evaluating alternatives based on weighted attributes [44]. In recent years, MCDM has been enhanced through integration with fuzzy logic systems, which allow for imprecise and qualitative information to be modeled, and through the adoption of behavioral decision-making theories—particularly bounded rationality, Prospect Theory, and Regret Theory.

Bounded Rationality, introduced by Herbert Simon [45], challenges the assumption that decision-makers are fully rational and possess perfect information. Instead, it acknowledges that decisions are made within cognitive, informational, and temporal constraints. In the context of international relations—where decisions often involve ambiguous information, long-term trade-offs, and competing strategic interests—bounded rationality offers a realistic framework for interpreting how countries assess partnerships like those with China [23,46,47]. This has given rise to models that incorporate behavioral responses to risk, uncertainty, and regret, rather than assuming purely utility-maximizing behavior.

Regret theory was developed by Graham Loomes and Robert Sugden in 1982 as a response to the limitations of expected utility theory, which assumes that people act solely to maximize their expected utility, without regard for potential emotional reactions to their decisions [48]. Regret theory extends this by considering that the utility of a current decision can be affected by comparisons to potential alternatives not chosen. Regret theory asserts that people anticipate the regret they might feel if the option they reject performs better than the option they chose, and the joy they might feel if their choice turns out to be better than the alternative. These emotions of regret and rejoicing are factored into the decision-making process, affecting the attractiveness of different choices.

To formalize regret theory, scholars introduce several key elements: $U(x)$ represents the utility derived from the outcome $x$, where $x$ is the outcome of the chosen action. Additionally, $y$ represents the outcome of an alternative action that was not chosen. The value function in regret theory is defined as:

$$V(x, y) = U(x) + R(x, y) \tag{1}$$

where $R(x, y)$ is the regret function. This function adjusts the utility of the chosen outcome $x$ by considering how it compares to the alternative outcome $y$. The regret function itself can be specified as follows:

$$R(x, y) = \begin{cases} -r\left(U(y) - U(x)\right) & \text{if } U(y) > U(x) \\ s\left(U(x) - U(y)\right) & \text{if } U(x) \geq U(y) \end{cases} \tag{2}$$

Here, $r$ and $s$ represent the coefficients of regret and rejoicing respectively. The function $R(x, y)$ is crucial because it reflects the psychological impact of comparing the chosen outcome to what could have been achieved with a different decision. If the non-chosen outcome $y$ is better than $x$, it generates regret proportional to the difference in their utilities, scaled by $r$. Conversely, if $x$ is better, it leads to rejoicing scaled by $s$.

Regret Theory, in particular, captures how decision-makers anticipate potential regret when faced with uncertain or conflicting options [21,48]. Its integration with MCDM methods such as MAIRCA, which compares observed outcomes to ideal benchmarks, enhances the ability to simulate real-world strategic behavior. Recent MCDM studies have embedded regret or prospect-based models in fields such as sustainable supply chains, investment decisions, and organizational performance evaluations. As summarized in Table 3, these models frequently leverage fuzzy sets or hybrid logic to represent ambiguous information, while incorporating psychological mechanisms like regret to more accurately reflect how decisions are actually made.

However, despite these advancements, most applications remain concentrated in business, engineering, or environmental domains, with relatively few addressing geopolitical or international development settings. The potential of combining regret-informed MCDM frameworks with international policy analysis remains underexplored. The current study seeks to address this gap by integrating LOPCOW, MAIRCA, and Regret Theory into a single behavioral-MCDM framework to analyze China–Africa relations—marking a novel application of these tools in a strategic geopolitical context.

Notation: Analytic Hierarchy Process (AHP), the Technique for Order Preference by Similarity to Ideal Solution (TOPSIS), the Full Consistency Method (FUCOM), the removal effects of criteria (MEREC), the Pythagorean Fuzzy Analytic Hierarchy Process (PFAHP), "VIekriterijumsko KOmpromisno Rangiranje" (VIKOR), Multi-Objective Optimiza- tion on the basis of a Ratio Analysis plus the full MULTIplicative form (MULTIMOORA), Decision Making Trial and Evaluation Laboratory (DEMATEL), TODIM (Portuguese acronym for Interactive Multi-Criteria Decision Making), DOmbi Bonferroni (DOBI)

## Methodology

### Objective and overview of the proposed framework

The primary methodological objective of this study is to develop and implement a regret theory-based MCDM framework to evaluate the level of Chinese influence in 49 African countries from 2018 to 2022. The model integrates the LOPCOW and MAIRCA techniques with Regret Theory, offering a behavioral decision-analytic lens to capture both the objective and subjective dimensions of strategic decision-making.

This study adopts a fourteen-step computational process that includes: normalization of data, transformation of utility under risk and regret behavior, calculation of objective weights using LOPCOW, and comparative performance ranking

**Table 3. The MCDM and bounded rationality studies.**

| Author(s) | Year | MCDM Method | Fuzzy Application | Bounded Rationality Theory |
|---|---|---|---|---|
| Fang R. & Liao H. [49] | 2021 | Evidential reasoning approach | No | Prospect Theory |
| Katarina R. et al. [50] | 2021 | Rough neutrosophic VIKOR | Rough neutrosophic sets | No |
| Zhao M. et al. [51] | 2022 | Personalized heuristic judgment MCDM | Yes | Prospect Theory |
| Tian C. et al. [52] | 2022 | Picture fuzzy MULTIMOORA | Picture fuzzy numbers | Prospect Theory |
| Pan X.-H. et al. [53] | 2022 | Interval type-2 fuzzy sets MCDM | Yes | Regret Theory |
| Le M. T. and Nhieu N-L. [54] | 2022 | Spherical fuzzy DEMATEL – TODIM | Spherical fuzzy sets | Prospect Theory |
| Ecer F. & Pamucar D. [55] | 2022 | Hybrid MCDM with LOPCOW and DOBI | No | No |
| Chai N. et al. [56] | 2023 | Fuzzy TOPSIS | Intuitionistic and interval-valued fuzzy sets | Cumulative Prospect Theory |
| Ding Q. et al. [57] | 2023 | DEMATEL | Interval sets | Regret Theory |
| Nila B. & Roy J. [58] | 2023 | Hybrid MCDM with LOPCOW, FUCOM, DOBI | Triangular fuzzy numbers | No |
| Esangbedo M. O. & Tang M. [59] | 2023 | Grey-MEREC-MAIRCA method | Grey system theory | No |
| Işık Ö., Çalık A., & Shabir M. [60] | 2024 | Hybrid MCDM with PFAHP and MAIRCA | Pythagorean fuzzy sets | No |
| *This study* | *2025* | *LOPCOW, MAIRCA in MCDM framework* | *No* | *Regret Theory* |

through MAIRCA. A central feature of the method is the use of gap analysis, which evaluates the distance between each country's actual strategic engagement and its theoretically ideal positioning, as defined by regret-adjusted utilities and weighted decision criteria. This approach enables both quantitative ranking and behavioral interpretation of influence.

**The proposed regret theory-based MCDM framework**

As illustrated in Fig 3, the proposed method follows these computational steps:

**Step 1.** Define Criteria and Alternatives:

Let $C = \{C_1, C_2, \ldots, C_n\}$ be the set of criteria.
  Let $A = \{A_1, A_2, \ldots, A_m\}$ be the set of alternatives.

**Step 2.** Collect Data:

Collect the decision matrix $X$ where each element $x_{ij}$ represents the value of criterion $C_j$ for alternative $A_i$.

**Step 3.** Normalization

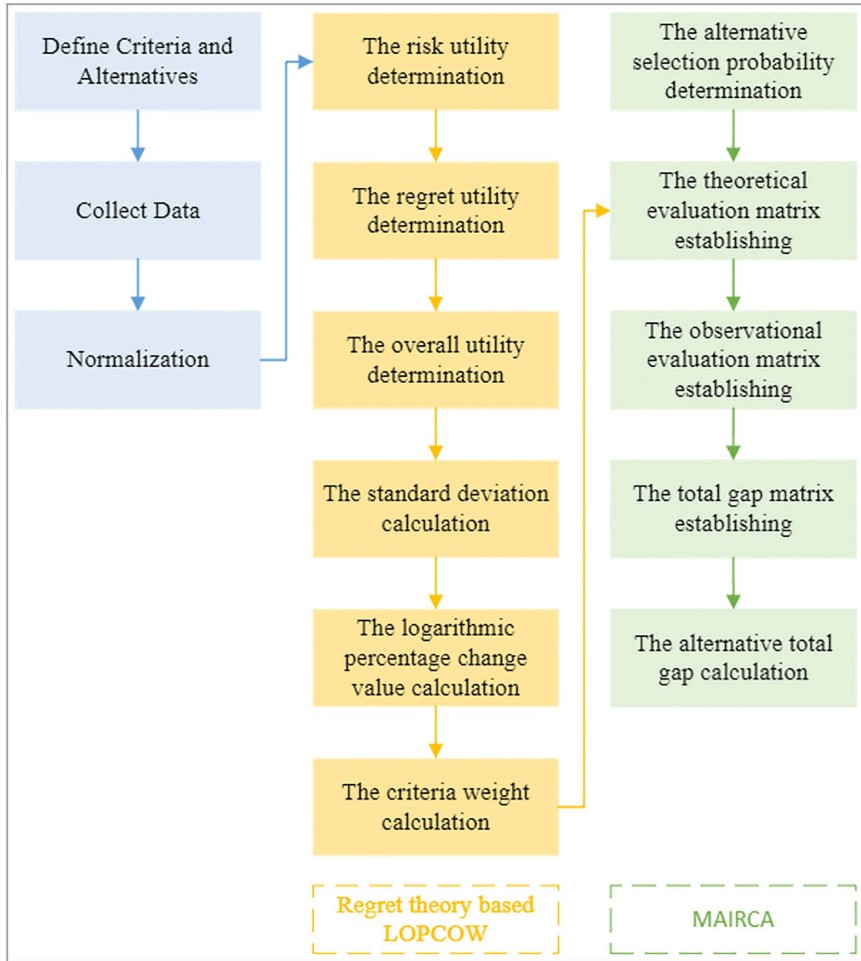

**Fig 3. The proposed approach.**

Normalize the decision matrix $X$ to get $X'$ by the min-max normalization method as:

$$x'_{ij} = \frac{x_{ij} - min(x_j)}{max(x_j) - min(x_j)} \quad \text{for benefit criteria}$$

(3)

$$x'_{ij} = \frac{max(x_j) - x_{ij}}{max(x_j) - min(x_j)} \quad \text{for cost criteria}$$

(4)

where $min(x_j)$ and $max(x_j)$ are the minimum and maximum values of the $j$-th criterion across all alternatives.

**Step 4.** The under-risk utility determination

Determine the under-risk utility of the normalized decision matrix based on the risk aversion coefficient $\alpha$ $(0 < \alpha \leq 1)$ of decision maker as follow. The lower value of $\alpha$, the smaller the decision maker's risk aversion.

$$U^{risk}\left(x'_{ij}\right) = \left(x'_{ij}\right)^{\alpha}$$

(5)

**Step 5.** The regret utility determination:

Determine the regret utility of normalized decision matrix based on the regret aversion coefficient $\beta$ $(0 < \beta \leq 1)$ of decision maker. The higher value of $\beta$, the greater the decision maker's regret aversion.

$$U^{regret}\left(x'_{ij}\right) = 1 - e^{-\beta\left(U^{risk}\left(x'_{ij}\right) - \max\limits_{j=1\ldots m}\left(U^{risk}\left(x'_{ij}\right)\right)\right)}$$

(6)

**Step 6.** The overall utility determination:

Determine the overall utility of normalized decision matrix by Equation (7). As the result, the regret theory decision matrix $Y$ is established as Equation (8).

$$U^{overall}\left(x'_{ij}\right) = U^{risk}\left(x'_{ij}\right) + U^{regret}\left(x'_{ij}\right)$$

(7)

$$Y = [y_{ij}]_{mxn} \quad \text{where } y_{ij} = U^{overall}\left(x'_{ij}\right) = U^{risk}$$

(8)

**Step 7.** The standard deviation calculation:

Calculate the standard deviation of each criterion in the regret theory decision matrix as:

$$s_j = \sqrt{\frac{\left(\sum_{i=1}^{m} y_{ij} - \bar{y}_j\right)^2}{m-1}} \quad \text{where } \bar{y}_j = \frac{\sum_{i=1}^{m} y_{ij}}{m}, \qquad j = 1\ldots n$$

(9)

**Step 8.** The logarithmic percentage change value calculation:

Calculate the logarithmic percentage change value ($L_{ij}$) of each criterion as Equation (10).

$$L_{ij} = \left| \ln\left(\frac{\sqrt{\frac{\sum_{i=1}^{m} y_{ij}^2}{m}}}{s_j}\right) \right|$$

(10)

 

**Step 9.** The criteria weight calculation:

Calculate the objective criteria weight ($w_j$) as Equation (11).

$$w_j = \frac{L_{ij}}{\sum_{i=1}^{m} L_{ij}}, \quad j = 1 \ldots n \tag{11}$$

**Step 10.** The alternative selection probability determination:

Determine the selection probability of alternatives. Usually, the decision maker provides equal probability to select the alternatives without bias. Therefore, their selection probability can be obtained as:

$$P_i = \frac{1}{m}, \quad i = 1 \ldots m \tag{12}$$

**Step 11.** The theoretical evaluation matrix establishing:

Establish the theoretical evaluation matrix by multiplying the alternative selection probability $P_i$ ($i = 1 \ldots m$) and the criteria weights $w_j$ ($j = 1 \ldots n$). Thus, the theoretical evaluation matrix $E^t$ is generated and shown as Equation (13).

$$E^t = \begin{bmatrix} e_{11}^t & \cdots & e_{1n}^t \\ \vdots & \ddots & \vdots \\ e_{m1}^t & \cdots & e_{mn}^t \end{bmatrix} = \begin{bmatrix} P_1 w_1 & \cdots & P_1 w_n \\ \vdots & \ddots & \vdots \\ P_m w_1 & \cdots & P_m w_n \end{bmatrix} \tag{13}$$

**Step 12.** The observational evaluation matrix establishing:

Establish the observational evaluation matrix as element-wise product of the normalized decision matrix ($X'$) and the theoretical evaluation matrix ($E^t$). Thus, the observational evaluation matrix ($E^{ob}$) is generated and shown as Equation (14).

$$E^{ob} = \begin{bmatrix} e_{11}^{ob} & \cdots & e_{1n}^{ob} \\ \vdots & \ddots & \vdots \\ e_{m1}^{ob} & \cdots & e_{mn}^{ob} \end{bmatrix} = \begin{bmatrix} e_{11}^t x_{11}' & \cdots & e_{1n}^t x_{1n}' \\ \vdots & \ddots & \vdots \\ e_{m1}^t x_{m1}' & \cdots & e_{mn}^t x_{mn}' \end{bmatrix} \tag{14}$$

**Step 13.** The total gap matrix establishing:

Establish the total gap matrix by element-wise subtraction of the theoretical evaluation matrix ($E^t$) and the observational evaluation matrix ($E^{ob}$). Thus, the total gap matrix is presented as Equation (15).

$$G = E^t - E^{ob} = \begin{bmatrix} g_{11} & \cdots & g_{1n} \\ \vdots & \ddots & \vdots \\ g_{m1} & \cdots & g_{mn} \end{bmatrix} = \begin{bmatrix} e_{11}^t - e_{11}^{ob} & \cdots & e_{1n}^t - e_{1n}^{ob} \\ \vdots & \ddots & \vdots \\ e_{m1}^t - e_{m1}^{ob} & \cdots & e_{mn}^t - e_{mn}^{ob} \end{bmatrix} \tag{15}$$

**Step 14.** The alternative total gap calculation:

The total gap of each alternative is calculated by Equation (16). The alternative with smaller total gap, the better alternative.

$$TG_i = \sum_{j=1}^{n} g_{ij}, \qquad i = 1 \ldots m$$

<div align="right">(16)</div>

## Ethical approval

This article does not contain any studies with human participants performed by any of the authors.

## Informed consent

This article does not contain any studies with human participants performed by any of the authors.

## Case study

**Problem descriptions.** In this section, the numerical results obtained from the analysis of Chinese influence in 49 African countries are presented as shown in Table 4. These countries were selected for analysis based on the availability and relevance of data regarding their engagements with Chinese. It should be noted that exclusion from this study of several African countries was necessitated by the absence of sufficient data to allow for a robust application of the proposed regret theory integrated MCDM approach. This extensive analysis employs the structured methodologies of LOPCOW and MAIRCA, combined with an assessment of potential regrets, to deliver insightful evaluations. The influence of Chinese engagements as perceived by these nations is meticulously examined, alongside the strategic decisions prompted by these engagements.

To thoroughly assess the multifaceted influence of Chinese in Africa, six critical indicators have been selected. Each of these indicators serves as a criterion within the proposed regret theory integrated MCDM approach to analyze economic interactions and their strategic implications.

- Chinese FDI Stock in African Countries (in million USD) – C1: The total value of Chinese foreign direct investment in Africa encapsulates Chinese economic entrenchment and its commitment to the continent. FDI is a vital indicator of direct economic influence, reflecting both the depth and breadth of Chinese economic engagements across diverse sectors such as mining, manufacturing, and infrastructure. These investments are not merely financial inputs but are strategic tools that can shape economic landscapes and policy orientations within African nations. A high FDI can suggest a greater dependency on Chinese capital, which could influence local economies and their development policies.

- Value of Chinese Exports to Africa by Country (in million USD) – C2: This indicator measures the economic scale of goods and services flowing from China to African nations, providing a clear picture of trade dynamics. Analyzing Chinese exports offers insights into how Chinese may be influencing African markets with its products, potentially leading to economic dependencies. Moreover, this flow represents a key element of Chinese export-driven growth strategy, where Africa serves as an important market for Chinese goods, thereby reinforcing economic ties.

- Value of Africa's Exports to China by Country (in million USD) – C3: The converse of Chinese exports, this measure reflects the value of exports from African countries to Chinese, highlighting critical export relationships. This indicator is essential for understanding how African economies are tethered to Chinese demand for raw materials and other goods. It sheds light on the economic balance and dependency on China as a major export destination, influencing local economies and their global economic positioning.

- Number of Chinese Labor (workers) – C4: The presence of Chinese workers in Africa is a direct reflection of Chinese involvement in local development projects. This labor migration is often associated with large-scale construction and mining projects, which are pivotal for infrastructure and industrial development. The number of Chinese workers can

**Table 4. The considered Africa countries.**

| No. | Country | Alpha-3 Code | Region | No. | Country | Alpha-3 Code | Region |
|---|---|---|---|---|---|---|---|
| 1 | Algeria | DZA | Northern Africa | 26 | Liberia | LBR | Western Africa |
| 2 | Angola | AGO | Central Africa | 27 | Madagascar | MDG | Eastern Africa |
| 3 | Benin | BEN | Western Africa | 28 | Malawi | MWI | Eastern Africa |
| 4 | Botswana | BWA | Southern Africa | 29 | Mali | MLI | Western Africa |
| 5 | Burkina Faso | BFA | Western Africa | 30 | Mauritania | MRT | Western Africa |
| 6 | Burundi | BDI | Eastern Africa | 31 | Mauritius | MUS | Eastern Africa |
| 7 | Cameroon | CMR | Central Africa | 32 | Morocco | MAR | Northern Africa |
| 8 | Cape Verde | CPV | Western Africa | 33 | Mozambique | MOZ | Eastern Africa |
| 9 | Central African Republic | CAF | Central Africa | 34 | Namibia | NAM | Southern Africa |
| 10 | Chad | TCD | Central Africa | 35 | Niger | NER | Western Africa |
| 11 | Comoros | COM | Eastern Africa | 36 | Nigeria | NGA | Western Africa |
| 12 | Republic of the Congo | COG | Central Africa | 37 | Rwanda | RWA | Eastern Africa |
| 13 | Democratic Republic of the Congo | COD | Central Africa | 38 | Senegal | SEN | Western Africa |
| 14 | Cote d'Ivoire | CIV | Western Africa | 39 | Seychelles | SYC | Eastern Africa |
| 15 | Djibouti | DJI | Eastern Africa | 40 | Sierra Leone | SLE | Western Africa |
| 16 | Egypt | EGY | Northern Africa | 41 | South Africa | ZAF | Southern Africa |
| 17 | Equatorial Guinea | GNQ | Central Africa | 42 | South Sudan | SSD | Eastern Africa |
| 18 | Eritrea | ERI | Eastern Africa | 43 | Sudan | SDN | Northern Africa |
| 19 | Ethiopia | ETH | Eastern Africa | 44 | Tanzania | TZA | Eastern Africa |
| 20 | Gabon | GAB | Central Africa | 45 | Togo | TGO | Western Africa |
| 21 | The Gambia | GMB | Western Africa | 46 | Tunisia | TUN | Northern Africa |
| 22 | Ghana | GHA | Western Africa | 47 | Uganda | UGA | Eastern Africa |
| 23 | Guinea | GIN | Western Africa | 48 | Zambia | ZMB | Eastern Africa |
| 24 | Kenya | KEN | Eastern Africa | 49 | Zimbabwe | ZWE | Eastern Africa |
| 25 | Lesotho | LSO | Southern Africa | | | | |

indicate the level of control and influence Chinese exerts over such projects and can have broader social and economic implications for the host countries, including impacts on local employment and skill development.

- Gross Annual Revenues of Chinese Companies' Construction Projects in Africa (in millions USD) – C5: This indicator provides a quantitative measure of the economic impact of Chinese construction activities in Africa. The revenues generated from these projects highlight the economic size and significance of Chinese involvement in Africa's infrastructure sector. Construction projects not only contribute to physical infrastructure but also play a strategic role in linking Chinese firms to local economies, potentially leading to increased influence and operational control.

- Loan from China (million USD) – C6: Financial loans from China to African nations are a critical aspect of Chinese strategy to expand its geopolitical influence through economic means. These loans often fund major infrastructure projects under the Belt and Road Initiative and other development projects, creating financial ties and potential dependencies. Analyzing these loans helps in understanding the fiscal leverage China holds over African countries, which can affect their economic sovereignty and decision-making autonomy.

## Data collection

The data collection for this study involves an extensive aggregation of information corresponding to six criteria across 49 African countries for the period 2018–2022. The sources of this data are highly credible and publicly accessible

databases, which include the SAIS China Africa Research Initiative [9], the International Labor Organization (ILO) [61], the International Monetary Fund (IMF) [62], the National Bureau of Statistics of China [63], and the World Trade Organization (WTO) [64]. Each of these organizations provides specific datasets that are critical for analyzing various aspects of Chinese influence in Africa, ranging from economic investments and trade flows to labor dynamics and financial loans. The 2022 collected data are presented in Table 5, while the 2018–2021 data are presented in Table A1-A4 in the Supplementary file.

## Numerical results

**Normalized matrix and utility matrices.** As described in the proposed regret theory-based MCDM framework section, before starting the procedures of the proposed approach, the decision matrix is normalized according to Equation (3)–(4). Accordingly, the normalized decision-making matrix for 2022 data was established and presented in Table 6, while the normalized decision matrices for remaining data are shown Table A5-A8 in the Supplementary file.

The subsequent phase involves the construction of matrices for under-risk utility, regret utility, and overall utility, in accordance with Equations (5) through (8), with the risk aversion coefficient ($\alpha$) and regret aversion coefficients ($\beta$) to their default values ($\alpha = 0.88$, $\beta = 0.3$) [47]. These matrices, delineating under-risk utility, regret utility, and overall utility, are presented comprehensively in Table 7–9 below.

## Standard deviation of regret-adjusted utilities

Using the overall utility matrix, the standard deviation of each criterion was computed to assess dispersion in influence-related variables across countries (Equation 9). These values are illustrated in Fig 4. The overall utility matrices of 2018–2021 data are shown in Tables A9-A12 in the Supplementary file.

## LOPCOW-based criteria weights

Logarithmic percentage change values ($L_{ij}$) were calculated for each criterion (Equation 10) and subsequently used to derive objective criteria weights ($w_j$) using the LOPCOW method (Equation 11). The results for 2022 are shown in Fig 5. Criteria weights from 2018–2021 were also computed and visualized in Fig 6, demonstrating temporal shifts in the relative importance of each indicator.

## Evaluation matrix, gap matrix construction

The probability of selecting each alternative is determined. In the absence of any bias, each country (alternative) is assigned an equal probability of being selected by the decision maker. Then, the theoretical evaluation matrix ($E^t$) and the observational evaluation matrix ($E^{ob}$) are established according to Equations (13)–(14). These matrices are shown in Tables A13-A22 in the Supplementary file. Then, the total gap matrix ($G$) is established, as shown in TableA23-A27 in the Supplementary file, by subtracting the observational evaluation matrix from the theoretical evaluation matrix on an element-wise basis according to Equation (15). Finally, the total gap for each alternative is calculated, which serves as a measure of each country's overall deviation from the theoretical model.

## Total gap calculation and influence rankings (2018–2022)

The total gap of an alternative ($TG_i$) is calculated by summing up the elements in its corresponding row within the gap matrix. Alternatives with smaller total gaps are considered better, as they are closer to the theoretical ideal. As shown in Fig 7, the 2022 data results reveal varying degrees of Chinese influence across different African countries, as indicated by the total gap values calculated from the regret theory-based MCDM analysis. Lower gap values denote a higher influence, suggesting that these countries' actual economic engagements with China closely align with the theoretical ideal model

**Table 5. The decision matrix for 2022 data.**

| Country | C1 | C2 | C3 | C4 | C5 | C6 |
|---|---|---|---|---|---|---|
| | million USD | million USD | million USD | workers | million USD | million USD |
| DZA | 1621.92 | 6276.10 | 1143.56 | 5371.00 | 2843.32 | 0.00 |
| AGO | 1946.17 | 4096.91 | 23245.99 | 3012.00 | 2887.16 | 18.57 |
| BEN | 168.62 | 1691.01 | 257.04 | 1102.00 | 669.10 | 39.32 |
| BWA | 143.43 | 221.34 | 398.38 | 298.00 | 487.98 | 0.00 |
| BFA | 6.64 | 504.26 | 104.02 | 322.00 | 74.79 | 0.00 |
| BDI | 19.59 | 119.95 | 10.07 | 153.00 | 28.17 | 0.00 |
| CMR | 389.68 | 3166.88 | 655.93 | 1700.00 | 659.96 | 0.00 |
| CPV | 1.62 | 92.96 | 0.02 | 12.00 | 1.68 | 0.00 |
| CAF | 9.58 | 52.45 | 25.03 | 142.00 | 50.06 | 0.00 |
| TCD | 567.76 | 282.06 | 1085.58 | 1101.00 | 658.87 | 0.00 |
| COM | 1.33 | 67.42 | 0.08 | 93.00 | 30.23 | 0.00 |
| COG | 395.22 | 976.34 | 5595.39 | 768.00 | 284.92 | 0.00 |
| COD | 4129.83 | 5117.59 | 16780.87 | 4579.00 | 2564.88 | 23.99 |
| CIV | 808.51 | 3490.90 | 970.22 | 2209.00 | 1566.14 | 199.54 |
| DJI | 85.82 | 3262.00 | 96.73 | 309.00 | 141.98 | 0.00 |
| EGY | 1203.37 | 17170.22 | 1019.82 | 6093.00 | 2734.86 | 0.00 |
| GNQ | 235.15 | 230.73 | 1516.38 | 588.00 | 368.74 | 0.00 |
| ERI | 320.43 | 148.27 | 461.79 | 478.00 | 69.78 | 0.00 |
| ETH | 2620.32 | 2216.94 | 453.60 | 3474.00 | 1584.14 | 0.00 |
| GAB | 152.16 | 583.25 | 3967.19 | 200.00 | 146.52 | 0.00 |
| GMB | 19.97 | 453.68 | 44.14 | 3.00 | 2.69 | 0.00 |
| GHA | 1058.26 | 7926.01 | 2344.01 | 1434.00 | 1289.75 | 84.99 |
| GIN | 1045.04 | 2283.33 | 4529.56 | 4213.00 | 1253.68 | 0.00 |
| KEN | 1782.42 | 8249.08 | 269.34 | 3351.00 | 2106.31 | 0.00 |
| LSO | 9.37 | 60.33 | 22.24 | 238.00 | 111.32 | 0.00 |
| LBR | 155.78 | 7520.36 | 15.00 | 273.00 | 131.16 | 0.00 |
| MDG | 281.94 | 1455.13 | 625.03 | 338.00 | 320.63 | 0.00 |
| MWI | 196.59 | 281.21 | 12.45 | 378.00 | 449.71 | 0.00 |
| MLI | 478.03 | 581.31 | 87.68 | 187.00 | 430.37 | 0.00 |
| MRT | 183.75 | 940.85 | 1187.14 | 273.00 | 45.37 | 0.00 |
| MUS | 1515.66 | 973.99 | 31.63 | 237.00 | 143.30 | 0.00 |
| MAR | 282.70 | 5741.10 | 909.51 | 138.00 | 242.17 | 0.00 |
| MOZ | 1180.35 | 3292.12 | 1340.12 | 959.00 | 751.71 | 0.00 |
| NAM | 176.92 | 556.85 | 593.50 | 315.00 | 286.87 | 0.00 |
| NER | 1853.56 | 675.91 | 317.71 | 1867.00 | 1015.69 | 0.00 |
| NGA | 2323.99 | 22299.68 | 1599.36 | 4478.00 | 4590.42 | 0.00 |
| RWA | 181.88 | 406.61 | 70.48 | 579.00 | 370.29 | 29.45 |
| SEN | 176.81 | 4068.33 | 269.51 | 1227.00 | 780.18 | 479.53 |
| SYC | 486.14 | 95.63 | 0.07 | 76.00 | 24.24 | 0.00 |
| SLE | 87.50 | 573.14 | 762.40 | 201.00 | 76.77 | 0.00 |
| ZAF | 5741.69 | 24196.44 | 11694.96 | 388.00 | 640.79 | 0.00 |
| SSD | 56.74 | 156.52 | 236.97 | 291.00 | 211.63 | 0.00 |
| SDN | 885.95 | 2033.57 | 877.30 | 777.00 | 343.93 | 0.00 |
| TZA | 1440.82 | 7774.68 | 535.80 | 2400.00 | 1301.81 | 0.00 |
| TGO | 55.99 | 3176.67 | 192.29 | 201.00 | 72.50 | 0.00 |

*(Continued)*

**Table 5.** (Continued)

| Country | C1 | C2 | C3 | C4 | C5 | C6 |
| --- | --- | --- | --- | --- | --- | --- |
| | million USD | million USD | million USD | workers | million USD | million USD |
| TUN | 26.20 | 1880.43 | 254.57 | 166.00 | 94.86 | 0.00 |
| UGA | 692.44 | 1076.90 | 58.28 | 1653.00 | 1108.98 | 119.10 |
| ZMB | 1979.57 | 979.68 | 5751.94 | 3007.00 | 1152.02 | 0.00 |
| ZWE | 1604.85 | 1124.59 | 1298.45 | 932.00 | 566.33 | 0.00 |

developed in this study. The rankings provided for African countries from 2018 to 2022, in relation to Chinese influence, display interesting trends and shifts over the five-year period as shown in Table 10. The total gap and ranking of Africa countries from 2018 to 2022 are shown in Table A28 in the supplementary file.

## Discussion

This study assessed the level and structure of Chinese influence in 49 African countries using an integrated regret theory-based MCDM framework combining LOPCOW and MAIRCA. The model quantified gaps between actual economic engagement and a regret-adjusted ideal, providing a nuanced behavioral view of strategic alignment.

### Interpretation of results

The results of the proposed regret theory-based MCDM framework reveal notable insights into the structural composition and geographical distribution of Chinese influence in Africa. The 2022 data showed that among the six criteria, the number of Chinese workers (C4) carried the highest weight (35.2%), followed by construction project revenues (C5) at 24.6%, and Chinese FDI stock (C1) at 18.4% (Fig 5). These findings indicate that influence is increasingly exercised through operational and project-based mechanisms—not just through capital inflow, but through the physical and human infrastructure embedded in African economies.

This represents a strategic shift in China's approach, reflecting a broader evolution from resource extraction and financial investment to labor presence, construction diplomacy, and project-based engagement. This trend is consistent with earlier findings, such as Dubinsky (2021) and Chiyemura et al. (2021), which document the expansion of China's overseas labor footprint and its alignment with an infrastructure-centric foreign policy across the Global South, particularly in Africa [39,41]. The labor presence reflects both China's domestic employment strategy and Africa's infrastructural needs. This supports earlier findings by Yang (2022), who argue that labor mobility is becoming an increasingly significant vector of geopolitical and economic influence [29]. In contrast, Chinese loans (C6) were assigned the lowest weight (2.3%), a significant departure from previous patterns of influence. This suggests that debt-based diplomacy, once a hallmark of China's Belt and Road engagements, may be becoming less central—either due to growing debt sustainability concerns among African countries or a recalibration of Chinese financing strategies. The variability of loan weights over the years (Fig 6), peaking in 2018 at 29.3% and falling to 1.0% in 2019, supports this interpretation, potentially indicating that major loan agreements were frontloaded early in the period and followed by a diversification of tactics.

The trend analysis from 2018 to 2022 further corroborates a structural transformation. While FDI (C1) was the dominant criterion in 2018 (24.1%), its importance declined steadily to 18.4% in 2022. Meanwhile, Chinese labor (C4) grew from only 2.4% in 2018 to the top weight of 35.2% by 2022 (Fig 6), implying a more embedded and operational mode of influence. This transition may reflect changing market conditions, African regulatory environments, or China's desire to transfer capabilities and manage projects directly rather than merely funding them, as discussed by Yang (2022) and Calabrese et al. (2023) [29,40].

**Table 6. The normalized decision matrix for 2022 data.**

| Country | C1 | C2 | C3 | C4 | C5 | C6 | Country | C1 | C2 | C3 | C4 | C5 | C6 |
|---------|------|------|------|------|------|------|---------|------|------|------|------|------|------|
| DZA | 0.282 | 0.258 | 0.049 | 0.881 | 0.619 | 0 | LBR | 0.027 | 0.309 | 0.001 | 0.044 | 0.028 | 0 |
| AGO | 0.339 | 0.168 | 1 | 0.494 | 0.629 | 0.039 | MDG | 0.049 | 0.058 | 0.027 | 0.055 | 0.07 | 0 |
| BEN | 0.029 | 0.068 | 0.011 | 0.18 | 0.145 | 0.082 | MWI | 0.034 | 0.009 | 0.001 | 0.062 | 0.098 | 0 |
| BWA | 0.025 | 0.007 | 0.017 | 0.048 | 0.106 | 0 | MLI | 0.083 | 0.022 | 0.004 | 0.03 | 0.093 | 0 |
| BFA | 0.001 | 0.019 | 0.004 | 0.052 | 0.016 | 0 | MRT | 0.032 | 0.037 | 0.051 | 0.044 | 0.01 | 0 |
| BDI | 0.003 | 0.003 | 0 | 0.025 | 0.006 | 0 | MUS | 0.264 | 0.038 | 0.001 | 0.038 | 0.031 | 0 |
| CMR | 0.068 | 0.129 | 0.028 | 0.279 | 0.143 | 0 | MAR | 0.049 | 0.236 | 0.039 | 0.022 | 0.052 | 0 |
| CPV | 0 | 0.002 | 0 | 0.001 | 0 | 0 | MOZ | 0.205 | 0.134 | 0.058 | 0.157 | 0.163 | 0 |
| CAF | 0.001 | 0 | 0.001 | 0.023 | 0.011 | 0 | NAM | 0.031 | 0.021 | 0.026 | 0.051 | 0.062 | 0 |
| TCD | 0.099 | 0.01 | 0.047 | 0.18 | 0.143 | 0 | NER | 0.323 | 0.026 | 0.014 | 0.306 | 0.221 | 0 |
| COM | 0 | 0.001 | 0 | 0.015 | 0.006 | 0 | NGA | 0.405 | 0.921 | 0.069 | 0.735 | 1 | 0 |
| COG | 0.069 | 0.038 | 0.241 | 0.126 | 0.062 | 0 | RWA | 0.031 | 0.015 | 0.003 | 0.095 | 0.08 | 0.061 |
| COD | 0.719 | 0.21 | 0.722 | 0.751 | 0.559 | 0.05 | SEN | 0.031 | 0.166 | 0.012 | 0.201 | 0.17 | 1 |
| CIV | 0.141 | 0.142 | 0.042 | 0.362 | 0.341 | 0.416 | SYC | 0.084 | 0.002 | 0 | 0.012 | 0.005 | 0 |
| DJI | 0.015 | 0.133 | 0.004 | 0.05 | 0.031 | 0 | SLE | 0.015 | 0.022 | 0.033 | 0.033 | 0.016 | 0 |
| EGY | 0.209 | 0.709 | 0.044 | 1 | 0.596 | 0 | ZAF | 1 | 1 | 0.503 | 0.063 | 0.139 | 0 |
| GNQ | 0.041 | 0.007 | 0.065 | 0.096 | 0.08 | 0 | SSD | 0.01 | 0.004 | 0.01 | 0.047 | 0.046 | 0 |
| ERI | 0.056 | 0.004 | 0.02 | 0.078 | 0.015 | 0 | SDN | 0.154 | 0.082 | 0.038 | 0.127 | 0.075 | 0 |
| ETH | 0.456 | 0.09 | 0.02 | 0.57 | 0.345 | 0 | TZA | 0.251 | 0.32 | 0.023 | 0.394 | 0.283 | 0 |
| GAB | 0.026 | 0.022 | 0.171 | 0.032 | 0.032 | 0 | TGO | 0.01 | 0.129 | 0.008 | 0.033 | 0.015 | 0 |
| GMB | 0.003 | 0.017 | 0.002 | 0 | 0 | 0 | TUN | 0.004 | 0.076 | 0.011 | 0.027 | 0.02 | 0 |
| GHA | 0.184 | 0.326 | 0.101 | 0.235 | 0.281 | 0.177 | UGA | 0.12 | 0.042 | 0.003 | 0.271 | 0.241 | 0.248 |
| GIN | 0.182 | 0.092 | 0.195 | 0.691 | 0.273 | 0 | ZMB | 0.345 | 0.038 | 0.247 | 0.493 | 0.251 | 0 |
| KEN | 0.31 | 0.339 | 0.012 | 0.55 | 0.459 | 0 | ZWE | 0.279 | 0.044 | 0.056 | 0.153 | 0.123 | 0 |
| LSO | 0.001 | 0 | 0.001 | 0.039 | 0.024 | 0 | | | | | | | |

In terms of country-level influence, the total gap analysis (Fig 7) shows that Nigeria (0.0061), Egypt (0.0075), DR Congo (0.0080), and Algeria (0.0092) are most aligned with the theoretical ideal model, suggesting that Chinese engagements in these countries are highly calibrated and multifaceted. These nations have long-standing economic relationships with China and are central to Chinese interests due to their natural resources, market size, political importance, or geo-strategic location [65,66]. Conversely, countries such as Cape Verde, Gambia, and Burundi consistently ranked at the bottom with the highest total gaps, indicating a limited alignment with the ideal model and suggesting either minimal Chinese engagement or a mismatch in strategic priorities. These countries may not feature prominently in China's geopolitical calculus due to their small economies, geographic isolation, or lower strategic leverage [6].

Finally, the year-by-year ranking trends (Table 10) confirm the durability of China's influence in certain key countries. Nigeria emerged as the top-ranked nation from 2019 to 2022, reflecting sustained bilateral engagement, possibly in sectors such as oil, transport, and digital infrastructure [67]. South Africa, which ranked highest in 2018, declined thereafter, which may reflect policy shifts, economic volatility, or the maturation of earlier investments. Meanwhile, countries such as DR Congo and Egypt demonstrated fluctuating but upward momentum, consistent with China's interests in energy and logistics, respectively.

These findings underscore the asymmetry and multidimensionality of China's influence in Africa—shaped not only by trade and investment, but also by labor deployment, infrastructure development, and the behavioral preferences of national actors as modeled through regret-based decision analysis.

**Table 7. The under-risk utility matrix for 2022 data.**

| Country | C1 | C2 | C3 | C4 | C5 | C6 | Country | C1 | C2 | C3 | C4 | C5 | C6 |
|---|---|---|---|---|---|---|---|---|---|---|---|---|---|
| DZA | 0.329 | 0.303 | 0.071 | 0.895 | 0.656 | 0 | LBR | 0.042 | 0.356 | 0.002 | 0.064 | 0.043 | 0 |
| AGO | 0.386 | 0.208 | 1 | 0.538 | 0.665 | 0.057 | MDG | 0.07 | 0.082 | 0.041 | 0.078 | 0.096 | 0 |
| BEN | 0.045 | 0.094 | 0.019 | 0.222 | 0.183 | 0.111 | MWI | 0.051 | 0.017 | 0.001 | 0.086 | 0.129 | 0 |
| BWA | 0.039 | 0.013 | 0.028 | 0.07 | 0.139 | 0 | MLI | 0.112 | 0.035 | 0.007 | 0.046 | 0.124 | 0 |
| BFA | 0.002 | 0.03 | 0.009 | 0.075 | 0.026 | 0 | MRT | 0.048 | 0.055 | 0.073 | 0.064 | 0.017 | 0 |
| BDI | 0.006 | 0.006 | 0.001 | 0.038 | 0.011 | 0 | MUS | 0.31 | 0.056 | 0.003 | 0.057 | 0.047 | 0 |
| CMR | 0.093 | 0.165 | 0.043 | 0.325 | 0.181 | 0 | MAR | 0.07 | 0.28 | 0.058 | 0.035 | 0.075 | 0 |
| CPV | 0 | 0.004 | 0 | 0.003 | 0 | 0 | MOZ | 0.248 | 0.171 | 0.081 | 0.196 | 0.203 | 0 |
| CAF | 0.003 | 0 | 0.002 | 0.036 | 0.018 | 0 | NAM | 0.046 | 0.033 | 0.04 | 0.073 | 0.087 | 0 |
| TCD | 0.13 | 0.017 | 0.067 | 0.221 | 0.181 | 0 | NER | 0.37 | 0.04 | 0.023 | 0.353 | 0.265 | 0 |
| COM | 0 | 0.002 | 0 | 0.025 | 0.011 | 0 | NGA | 0.451 | 0.931 | 0.095 | 0.762 | 1 | 0 |
| COG | 0.095 | 0.057 | 0.286 | 0.161 | 0.086 | 0 | RWA | 0.048 | 0.024 | 0.006 | 0.126 | 0.109 | 0.086 |
| COD | 0.748 | 0.253 | 0.751 | 0.778 | 0.599 | 0.072 | SEN | 0.046 | 0.206 | 0.02 | 0.244 | 0.21 | 1 |
| CIV | 0.178 | 0.18 | 0.061 | 0.409 | 0.388 | 0.462 | SYC | 0.114 | 0.004 | 0 | 0.02 | 0.009 | 0 |
| DJI | 0.024 | 0.169 | 0.008 | 0.072 | 0.046 | 0 | SLE | 0.025 | 0.034 | 0.049 | 0.049 | 0.027 | 0 |
| EGY | 0.253 | 0.739 | 0.064 | 1 | 0.634 | 0 | ZAF | 1 | 1 | 0.546 | 0.088 | 0.176 | 0 |
| GNQ | 0.06 | 0.013 | 0.091 | 0.127 | 0.108 | 0 | SSD | 0.017 | 0.008 | 0.018 | 0.068 | 0.066 | 0 |
| ERI | 0.079 | 0.008 | 0.032 | 0.106 | 0.025 | 0 | SDN | 0.193 | 0.111 | 0.056 | 0.163 | 0.102 | 0 |
| ETH | 0.501 | 0.12 | 0.031 | 0.61 | 0.392 | 0 | TZA | 0.296 | 0.367 | 0.036 | 0.44 | 0.33 | 0 |
| GAB | 0.041 | 0.035 | 0.211 | 0.049 | 0.048 | 0 | TGO | 0.017 | 0.165 | 0.015 | 0.049 | 0.025 | 0 |
| GMB | 0.006 | 0.027 | 0.004 | 0 | 0.001 | 0 | TUN | 0.008 | 0.103 | 0.019 | 0.041 | 0.032 | 0 |
| GHA | 0.226 | 0.373 | 0.133 | 0.28 | 0.327 | 0.218 | UGA | 0.155 | 0.062 | 0.005 | 0.317 | 0.286 | 0.294 |
| GIN | 0.223 | 0.123 | 0.237 | 0.723 | 0.319 | 0 | ZMB | 0.392 | 0.057 | 0.293 | 0.537 | 0.296 | 0 |
| KEN | 0.357 | 0.386 | 0.02 | 0.591 | 0.504 | 0 | ZWE | 0.326 | 0.065 | 0.079 | 0.191 | 0.158 | 0 |
| LSO | 0.003 | 0.001 | 0.002 | 0.057 | 0.037 | 0 |  |  |  |  |  |  |  |

## Theoretical contribution and novelty

The novelty of this study lies in the integration of regret theory with a multi-criteria decision-making framework (LOP-COW–MAIRCA) to assess geopolitical influence—a methodology not previously applied to China–Africa relations. While prior studies have examined trade, FDI, and soft power individually, this study offers a behaviorally grounded, data-driven model capable of simulating both rational trade-offs and bounded decision behavior. It extends the literature by introducing regret-adjusted gap analysis into strategic assessment, moving beyond descriptive statistics to capture preference reversals and uncertainty-based reasoning common in international policymaking.

## Policy implications

The findings of this study offer actionable insights for governments, regional bodies, and international development partners seeking to navigate the evolving landscape of China–Africa relations. The regret theory-based MCDM framework enables stakeholders to move beyond surface-level indicators and assess how closely national engagement aligns with strategic and behavioral optima.

First, the model provides a benchmarking tool. By calculating total gap scores and influence rankings, countries can compare their current positioning relative to an ideal strategic configuration. This allows decision-makers to assess whether their economic relationship with China is underperforming, balanced, or over-aligned based on multiple dimensions such as labor, investment, trade, and loans.

 

**Table 8. The regret utility matrix for 2022 data.**

| Country | C1 | C2 | C3 | C4 | C5 | C6 | Country | C1 | C2 | C3 | C4 | C5 | C6 |
|---|---|---|---|---|---|---|---|---|---|---|---|---|---|
| DZA | −0.223 | −0.232 | −0.322 | −0.032 | −0.109 | −0.35 | LBR | −0.333 | −0.213 | −0.349 | −0.324 | −0.332 | −0.35 |
| AGO | −0.202 | −0.268 | 0 | −0.149 | −0.106 | −0.327 | MDG | −0.322 | −0.317 | −0.333 | −0.319 | −0.312 | −0.35 |
| BEN | −0.332 | −0.312 | −0.342 | −0.263 | −0.278 | −0.306 | MWI | −0.329 | −0.343 | −0.349 | −0.315 | −0.299 | −0.35 |
| BWA | −0.334 | −0.345 | −0.339 | −0.322 | −0.295 | −0.35 | MLI | −0.305 | −0.336 | −0.347 | −0.331 | −0.301 | −0.35 |
| BFA | −0.349 | −0.338 | −0.346 | −0.32 | −0.339 | −0.35 | MRT | −0.331 | −0.328 | −0.321 | −0.324 | −0.343 | −0.35 |
| BDI | −0.347 | −0.348 | −0.349 | −0.334 | −0.346 | −0.35 | MUS | −0.23 | −0.327 | −0.349 | −0.327 | −0.331 | −0.35 |
| CMR | −0.313 | −0.285 | −0.332 | −0.225 | −0.278 | −0.35 | MAR | −0.322 | −0.241 | −0.327 | −0.336 | −0.32 | −0.35 |
| CPV | −0.35 | −0.348 | −0.35 | −0.349 | −0.35 | −0.35 | MOZ | −0.253 | −0.282 | −0.317 | −0.273 | −0.27 | −0.35 |
| CAF | −0.349 | −0.35 | −0.349 | −0.335 | −0.343 | −0.35 | NAM | −0.331 | −0.336 | −0.334 | −0.321 | −0.315 | −0.35 |
| TCD | −0.298 | −0.343 | −0.323 | −0.263 | −0.279 | −0.35 | NER | −0.208 | −0.334 | −0.341 | −0.214 | −0.247 | −0.35 |
| COM | −0.35 | −0.349 | −0.35 | −0.34 | −0.345 | −0.35 | NGA | −0.179 | −0.021 | −0.312 | −0.074 | 0 | −0.35 |
| COG | −0.312 | −0.327 | −0.239 | −0.286 | −0.315 | −0.35 | RWA | −0.331 | −0.34 | −0.347 | −0.3 | −0.307 | −0.316 |
| COD | −0.078 | −0.251 | −0.078 | −0.069 | −0.128 | −0.321 | SEN | −0.331 | −0.269 | −0.342 | −0.255 | −0.267 | 0 |
| CIV | −0.28 | −0.279 | −0.325 | −0.194 | −0.202 | −0.175 | SYC | −0.305 | −0.348 | −0.35 | −0.342 | −0.346 | −0.35 |
| DJI | −0.34 | −0.283 | −0.347 | −0.321 | −0.331 | −0.35 | SLE | −0.34 | −0.336 | −0.33 | −0.33 | −0.339 | −0.35 |
| EGY | −0.251 | −0.081 | −0.324 | 0 | −0.116 | −0.35 | ZAF | 0 | 0 | −0.146 | −0.315 | −0.28 | −0.35 |
| GNQ | −0.326 | −0.344 | −0.314 | −0.299 | −0.307 | −0.35 | SSD | −0.343 | −0.347 | −0.343 | −0.323 | −0.323 | −0.35 |
| ERI | −0.318 | −0.347 | −0.337 | −0.308 | −0.34 | −0.35 | SDN | −0.274 | −0.306 | −0.327 | −0.286 | −0.309 | −0.35 |
| ETH | −0.161 | −0.302 | −0.337 | −0.124 | −0.2 | −0.35 | TZA | −0.235 | −0.209 | −0.335 | −0.183 | −0.223 | −0.35 |
| GAB | −0.333 | −0.336 | −0.267 | −0.33 | −0.331 | −0.35 | TGO | −0.343 | −0.285 | −0.344 | −0.33 | −0.34 | −0.35 |
| GMB | −0.347 | −0.339 | −0.348 | −0.35 | −0.35 | −0.35 | TUN | −0.346 | −0.309 | −0.342 | −0.333 | −0.337 | −0.35 |
| GHA | −0.262 | −0.207 | −0.297 | −0.241 | −0.224 | −0.264 | UGA | −0.288 | −0.325 | −0.348 | −0.227 | −0.239 | −0.236 |
| GIN | −0.262 | −0.301 | −0.257 | −0.087 | −0.227 | −0.35 | ZMB | −0.2 | −0.327 | −0.236 | −0.149 | −0.235 | −0.35 |
| KEN | −0.213 | −0.202 | −0.342 | −0.131 | −0.161 | −0.35 | ZWE | −0.224 | −0.324 | −0.318 | −0.275 | −0.287 | −0.35 |
| LSO | −0.349 | −0.35 | −0.349 | −0.327 | −0.335 | −0.35 | | | | | | | |

Second, the methodology supports the identification of overexposure or strategic imbalance. For example, countries heavily reliant on loans (previously dominant in 2018) may reconsider financial dependence in light of recent shifts in Chinese influence patterns. Similarly, disproportionate reliance on labor-intensive engagement or construction projects could expose vulnerabilities related to employment displacement, sovereignty, or limited local capacity building.

Third, the framework aids in informing national and regional policy decisions. Governments can use the criteria weight trends and influence alignment to inform diversification strategies, strengthen negotiation positions, and design policies that better align Chinese engagement with long-term development goals, including infrastructure resilience, workforce development, and trade competitiveness. International stakeholders and development finance institutions can also leverage these insights to support more balanced partnerships that complement rather than duplicate Chinese initiatives.

## Limitations

Despite the novel methodological contributions and insights presented in this study, several limitations must be acknowledged that frame the interpretation and generalizability of the findings.

First and foremost, the study is constrained by the availability, coverage, and transparency of data used to construct the indicators of Chinese influence. While six key indicators were selected to provide a multidimensional perspective—covering investment, trade, labor, construction revenues, and loans—data on China-Africa relations are often opaque, inconsistently reported, or subject to political filtering. Many bilateral deals, particularly in infrastructure and financing, occur

**Table 9. The overall utility matrix for 2022 data.**

| Country | C1 | C2 | C3 | C4 | C5 | C6 | Country | C1 | C2 | C3 | C4 | C5 | C6 |
|---|---|---|---|---|---|---|---|---|---|---|---|---|---|
| DZA | 0.105 | 0.071 | 0 | 0.863 | 0.547 | 0 | LBR | 0 | 0.143 | 0 | 0 | 0 | 0 |
| AGO | 0.183 | 0 | 1 | 0.389 | 0.559 | 0 | MDG | 0 | 0 | 0 | 0 | 0 | 0 |
| BEN | 0 | 0 | 0 | 0 | 0 | 0 | MWI | 0 | 0 | 0 | 0 | 0 | 0 |
| BWA | 0 | 0 | 0 | 0 | 0 | 0 | MLI | 0 | 0 | 0 | 0 | 0 | 0 |
| BFA | 0 | 0 | 0 | 0 | 0 | 0 | MRT | 0 | 0 | 0 | 0 | 0 | 0 |
| BDI | 0 | 0 | 0 | 0 | 0 | 0 | MUS | 0.079 | 0 | 0 | 0 | 0 | 0 |
| CMR | 0 | 0 | 0 | 0.1 | 0 | 0 | MAR | 0 | 0.039 | 0 | 0 | 0 | 0 |
| CPV | 0 | 0 | 0 | 0 | 0 | 0 | MOZ | 0 | 0 | 0 | 0 | 0 | 0 |
| CAF | 0 | 0 | 0 | 0 | 0 | 0 | NAM | 0 | 0 | 0 | 0 | 0 | 0 |
| TCD | 0 | 0 | 0 | 0 | 0 | 0 | NER | 0.161 | 0 | 0 | 0.139 | 0.018 | 0 |
| COM | 0 | 0 | 0 | 0 | 0 | 0 | NGA | 0.272 | 0.909 | 0 | 0.689 | 1 | 0 |
| COG | 0 | 0 | 0.047 | 0 | 0 | 0 | RWA | 0 | 0 | 0 | 0 | 0 | 0 |
| COD | 0.67 | 0.002 | 0.673 | 0.709 | 0.471 | 0 | SEN | 0 | 0 | 0 | 0 | 0 | 1 |
| CIV | 0 | 0 | 0 | 0.215 | 0.186 | 0.287 | SYC | 0 | 0 | 0 | 0 | 0 | 0 |
| DJI | 0 | 0 | 0 | 0 | 0 | 0 | SLE | 0 | 0 | 0 | 0 | 0 | 0 |
| EGY | 0.001 | 0.657 | 0 | 1 | 0.518 | 0 | ZAF | 1 | 1 | 0.401 | 0 | 0 | 0 |
| GNQ | 0 | 0 | 0 | 0 | 0 | 0 | SSD | 0 | 0 | 0 | 0 | 0 | 0 |
| ERI | 0 | 0 | 0 | 0 | 0 | 0 | SDN | 0 | 0 | 0 | 0 | 0 | 0 |
| ETH | 0.34 | 0 | 0 | 0.486 | 0.192 | 0 | TZA | 0.061 | 0.157 | 0 | 0.257 | 0.107 | 0 |
| GAB | 0 | 0 | 0 | 0 | 0 | 0 | TGO | 0 | 0 | 0 | 0 | 0 | 0 |
| GMB | 0 | 0 | 0 | 0 | 0 | 0 | TUN | 0 | 0 | 0 | 0 | 0 | 0 |
| GHA | 0 | 0.166 | 0 | 0.038 | 0.103 | 0 | UGA | 0 | 0 | 0 | 0.089 | 0.047 | 0.057 |
| GIN | 0 | 0 | 0 | 0.636 | 0.092 | 0 | ZMB | 0.191 | 0 | 0.056 | 0.388 | 0.061 | 0 |
| KEN | 0.144 | 0.184 | 0 | 0.46 | 0.343 | 0 | ZWE | 0.101 | 0 | 0 | 0 | 0 | 0 |
| LSO | 0 | 0 | 0 | 0 | 0 | 0 | | | | | | | |

outside the purview of standardized international reporting systems. This can result in underreporting or overaggregation, especially in countries with weak statistical institutions or limited public disclosure norms. Consequently, certain aspects of China's influence may have been overlooked or inadequately captured due to data limitations.

Second, although the chosen indicators represent distinct dimensions of economic and operational engagement, they do not capture soft power or non-economic channels such as cultural diplomacy, education exchanges, media influence, political alignment, or military cooperation. These are increasingly relevant in discussions on geopolitical influence but remain difficult to quantify. Furthermore, the exclusive reliance on quantitative metrics limits the ability to incorporate qualitative nuances, such as local perceptions of Chinese involvement or the political context within recipient countries. Future studies should consider integrating qualitative interviews or case-specific narratives to complement the empirical findings and better reflect the socio-political complexity of influence.

Third, the regret theory-based multi-criteria decision-making (MCDM) framework, while offering a valuable behavioral extension to traditional utility-based models, is still a stylized representation of actual decision-making processes at the national level. Governments do not always act in a manner consistent with formal rational choice or utility optimization frameworks. Instead, decisions are often driven by bureaucratic politics, historical ties, domestic interests, or ideological orientations. While the model integrates psychological constructs such as regret aversion and risk sensitivity, these parameters are applied uniformly across all countries, which may oversimplify the heterogeneity of national behaviors and institutional settings.

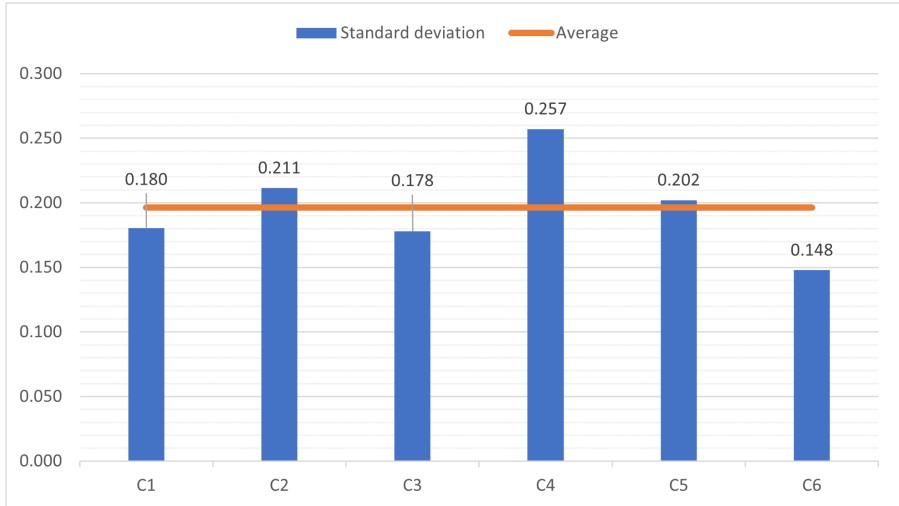

**Fig 4. The standard deviation of criteria for 2022 data.**

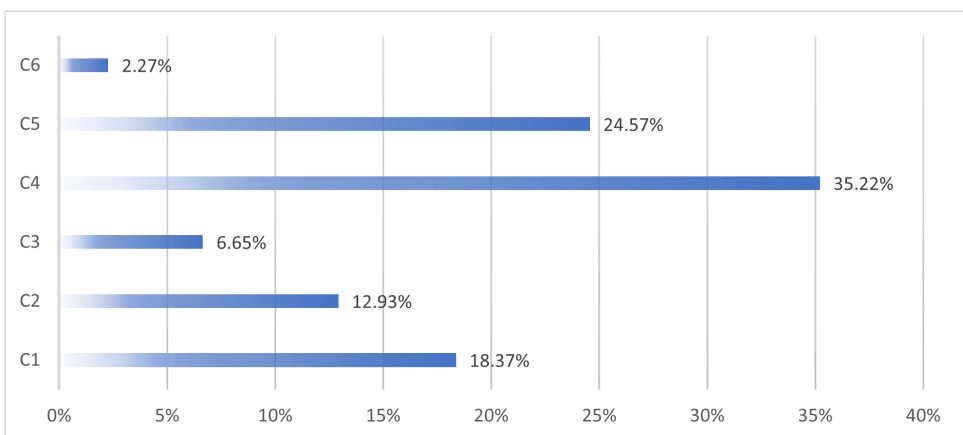

**Fig 5. The objective criteria weight for 2022 data.**

Fourth, the model assumes equal decision-making probabilities for all alternatives (countries) in the observational evaluation phase, which may not accurately reflect the geopolitical or economic priority of each country from China's perspective. While this assumption ensures neutrality in model construction, it may obscure strategic differentiation that China applies in practice based on natural resource endowments, regional security interests, or diplomatic alignments. Future refinements could consider incorporating differentiated weighting schemes or endogenous prioritization criteria derived from foreign policy analysis.

Fifth, the temporal structure of the analysis, while including multiple years (2018–2022), is essentially cross-sectional on a yearly basis and does not incorporate time-series modeling or longitudinal causal inference. As a result, the study cannot fully account for lagged effects, feedback loops, or shocks such as COVID-19, which may have disrupted trade, labor mobility, and financing patterns in ways not immediately reflected in annual data. Future research should explore

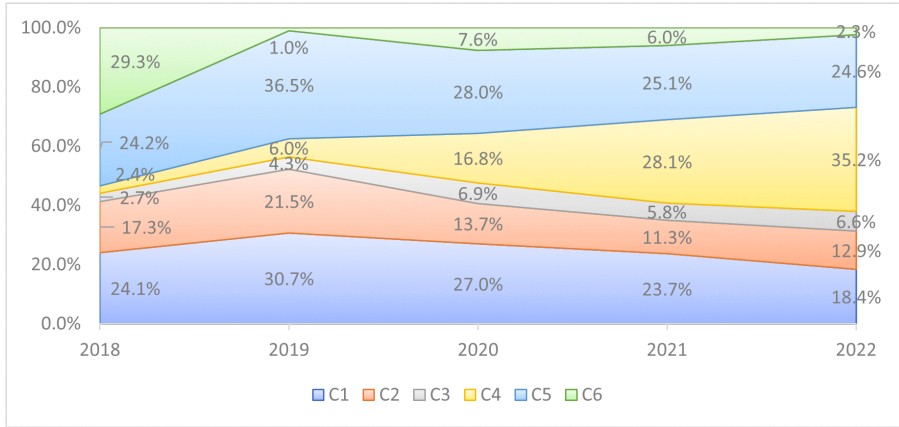

**Fig 6. The objective criteria weight.**

dynamic panel approaches, structural break analysis, or path dependency models to uncover deeper temporal patterns and their implications for influence trajectories.

Lastly, the model does not explicitly assess the consequences of Chinese influence on developmental outcomes in recipient countries. While it ranks and quantifies relative influence, the implications—whether positive, negative, or mixed—for economic sovereignty, governance quality, environmental sustainability, or local employment are not explored. A more comprehensive policy evaluation framework could be developed to link influence metrics to long-term socioeconomic impacts, helping governments and stakeholders distinguish between benign partnerships and potential dependencies or risks.

## Conclusion

This study was motivated by the expanding footprint of Chinese in Africa and the need for a nuanced understanding of this influence. The significance of Africa's strategic position and resources, coupled with Chinese global ambitions, has led to a proliferation of economic and political engagements across the continent. The motivation behind this research was to dissect these interactions through a methodological lens that accounts for both the quantitative and psychological dimensions of decision-making.

Employing a regret theory-based MCDM approach, the study methodically evaluated Chinese influence across 49 African countries using six economic criteria from 2018 to 2022. The methodology involved normalization, determination of regret utility, and calculation of total gap values, culminating in a ranking system that delineates the intensity of Chinese influence within each country.

The study revealed that while some countries exhibit a growing alignment with Chinese economic strategies, others maintain a more distant or fluctuating relationship. Key findings include significant year-on-year shifts in Chinese influence, with nations like Nigeria and Egypt showcasing heightened levels of engagement, whereas others like Cape Verde reflected lesser influence. The findings also highlighted sectors of Chinese focus, such as infrastructure and labor, and indicated varying degrees of trade relationships.

The study makes several contributions to the existing body of knowledge. It introduces a novel application of regret theory in geopolitical analysis, offers a detailed empirical assessment of Chinese influence in Africa, and provides a decision-making framework that African policymakers can use to gauge and strategize their engagements with Chinese. Furthermore, it enriches the dialogue on international relations by providing a model that blends economic data with behavioral economics.

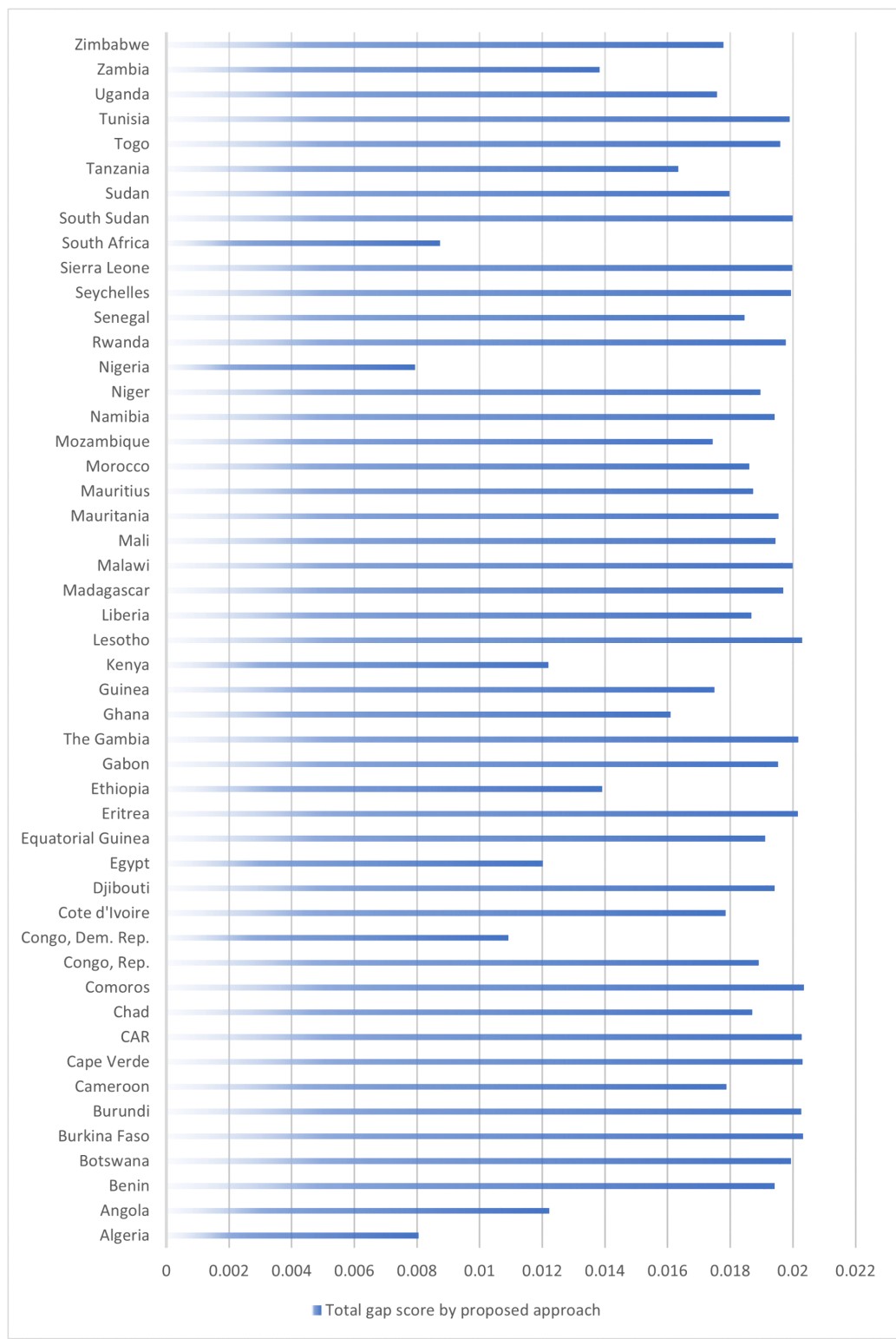

**Fig 7. The total gap score (TG$_i$) of Africa countries in 2022.**

**Table 10. The ranking according to Chinese influence from 2018 to 2022.**

| Country | 2018 | 2019 | 2020 | 2021 | 2022 |
|---|---|---|---|---|---|
| DZA | 5 | 2 | 1 | 2 | 4 |
| AGO | 4 | 7 | 7 | 4 | 5 |
| BEN | 34 | 27 | 35 | 23 | 21 |
| BWA | 38 | 37 | 38 | 33 | 32 |
| BFA | 47 | 48 | 44 | 46 | 41 |
| BDI | 48 | 44 | 46 | 44 | 45 |
| CMR | 17 | 17 | 18 | 18 | 16 |
| CPV | 49 | 47 | 49 | 49 | 49 |
| CAF | 46 | 45 | 47 | 45 | 46 |
| TCD | 27 | 22 | 21 | 21 | 20 |
| COM | 43 | 49 | 48 | 48 | 47 |
| COG | 22 | 24 | 17 | 20 | 23 |
| COD | 8 | 4 | 4 | 3 | 3 |
| CIV | 15 | 16 | 16 | 14 | 12 |
| DJI | 29 | 28 | 33 | 35 | 33 |
| EGY | 2 | 5 | 8 | 7 | 2 |
| GNQ | 26 | 26 | 26 | 28 | 26 |
| ERI | 37 | 42 | 43 | 42 | 35 |
| ETH | 9 | 9 | 5 | 5 | 8 |
| GAB | 35 | 31 | 31 | 34 | 36 |
| GMB | 45 | 43 | 45 | 47 | 48 |
| GHA | 11 | 10 | 10 | 12 | 13 |
| GIN | 7 | 13 | 12 | 11 | 9 |
| KEN | 10 | 6 | 6 | 8 | 6 |
| LSO | 41 | 46 | 42 | 43 | 44 |
| LBR | 28 | 21 | 27 | 30 | 25 |
| MDG | 21 | 34 | 28 | 25 | 29 |
| MWI | 42 | 41 | 32 | 38 | 30 |
| MLI | 24 | 30 | 29 | 27 | 31 |
| MRT | 36 | 32 | 34 | 36 | 38 |
| MUS | 25 | 23 | 25 | 24 | 24 |
| MAR | 19 | 20 | 23 | 26 | 27 |
| MOZ | 18 | 12 | 14 | 16 | 18 |
| NAM | 30 | 29 | 30 | 31 | 34 |
| NER | 31 | 25 | 20 | 19 | 14 |
| NGA | 3 | 1 | 2 | 1 | 1 |
| RWA | 32 | 35 | 24 | 29 | 28 |
| SEN | 12 | 19 | 22 | 15 | 17 |
| SYC | 40 | 38 | 39 | 39 | 43 |
| SLE | 44 | 39 | 41 | 41 | 42 |
| ZAF | 1 | 3 | 3 | 6 | 7 |
| SSD | 23 | 40 | 37 | 32 | 39 |
| SDN | 20 | 18 | 19 | 22 | 22 |
| TZA | 13 | 11 | 11 | 10 | 11 |
| TGO | 33 | 33 | 36 | 37 | 37 |
| TUN | 39 | 36 | 40 | 40 | 40 |

*(Continued)*

**Table 10.** (Continued)

| Country | 2018 | 2019 | 2020 | 2021 | 2022 |
|---------|------|------|------|------|------|
| UGA | 14 | 14 | 13 | 13 | 15 |
| ZMB | 6 | 8 | 9 | 9 | 10 |
| ZWE | 16 | 15 | 15 | 17 | 19 |

In light of these limitations, future research is encouraged to: (1) broaden the scope of indicators to include soft power and political dimensions, (2) incorporate qualitative data from expert panels or country-specific fieldwork, (3) introduce dynamic modeling to trace the evolution of influence, and (4) evaluate the developmental outcomes associated with varying levels of Chinese engagement. These steps will contribute to a more robust, policy-relevant, and contextually grounded understanding of how global powers like China shape regional development pathways through multifaceted engagement strategies.

## Supporting information

**S1 Data. Minimal data.**
(DOCX)

## Author contributions

**Conceptualization:** Chia-Nan Wang, Nhat-Luong Nhieu, Ching-Ju Lu.

**Data curation:** Nhat-Luong Nhieu.

**Formal analysis:** Chia-Nan Wang, Nhat-Luong Nhieu, Ching-Ju Lu.

**Funding acquisition:** Nhat-Luong Nhieu, Ching-Ju Lu.

**Investigation:** Nhat-Luong Nhieu.

**Methodology:** Nhat-Luong Nhieu.

**Project administration:** Nhat-Luong Nhieu, Ching-Ju Lu.

**Resources:** Nhat-Luong Nhieu.

**Software:** Nhat-Luong Nhieu, Ching-Ju Lu.

**Validation:** Nhat-Luong Nhieu.

**Visualization:** Nhat-Luong Nhieu.

**Writing – original draft:** Nhat-Luong Nhieu, Ching-Ju Lu.

**Writing – review & editing:** Chia-Nan Wang, Nhat-Luong Nhieu, Ching-Ju Lu.

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
