## [Decision Letter · Decision Letter 0]

4 Apr 2025

Dear Dr. Nhieu,

Thank you for submitting your manuscript to PLOS ONE. After careful consideration, we feel that it has merit but does not fully meet PLOS ONE’s publication criteria as it currently stands. Therefore, we invite you to submit a revised version of the manuscript that addresses the points raised during the review process.

We look forward to receiving your revised manuscript.

Kind regards,

Pasquale Marcello Falcone

Academic Editor

PLOS ONE

Journal Requirements:

This research is partial funded by University of Economics Ho Chi Minh City, VietNam

4. We note that your Data Availability Statement is currently as follows: All relevant data are within the manuscript and its Supporting Information files

5. We note that Figure 7 in your submission contain [map/satellite] images which may be copyrighted. All PLOS content is published under the Creative Commons Attribution License (CC BY 4.0), which means that the manuscript, images, and Supporting Information files will be freely available online, and any third party is permitted to access, download, copy, distribute, and use these materials in any way, even commercially, with proper attribution. For these reasons, we cannot publish previously copyrighted maps or satellite images created using proprietary data, such as Google software (Google Maps, Street View, and Earth). For more information, see our copyright guidelines: http://journals.plos.org/plosone/s/licenses-and-copyright.

a. You may seek permission from the original copyright holder of Figure 7 to publish the content specifically under the CC BY 4.0 license.

6. Please include a copy of Table A1-A4, A5-A8, A9-A12, A13-A22, A23-A27 and A28 which you refer to in your text on page 11, 12, 13, 16, and 18.

Additional Editor Comments :

Thank you for your submission. The manuscript presents an ambitious attempt to quantify Chinese influence in Africa using regret theory and decision-making approaches. While the study's motivation is clear and relevant, significant concerns have been raised regarding its conceptual, methodological, and empirical foundations.

One reviewer has recommended rejection, citing the lack of strong theoretical justification for the chosen methods, issues with data completeness and verification, and concerns about the robustness of the methodology in assessing geopolitical influence. However, the paper’s core research question is deemed valuable, and with substantial improvements—particularly in refining the theoretical framework, strengthening methodological rigor, and ensuring empirical validation—the study has potential for contribution.

We invite the authors to undertake a major revision addressing these concerns. In particular, the theoretical justification for applying regret theory and MCDM methods must be significantly strengthened, the data sources must be clarified and verified, and the overall methodological approach must be made more robust to meaningfully assess Chinese influence. A clearer definition and operationalization of "influence" would also enhance the study’s impact.

Please provide a detailed response to the reviewers’ comments, outlining how each concern has been addressed. We look forward to reviewing a revised version of the manuscript.

Reviewers' comments:

Reviewer's Responses to Questions

**Comments to the Author**

1. Is the manuscript technically sound, and do the data support the conclusions?

Reviewer #1: No

Reviewer #2: Yes

2. Has the statistical analysis been performed appropriately and rigorously?

Reviewer #1: No

Reviewer #2: I Don't Know

3. Have the authors made all data underlying the findings in their manuscript fully available?

Reviewer #1: No

Reviewer #2: No

4. Is the manuscript presented in an intelligible fashion and written in standard English?

Reviewer #1: No

Reviewer #2: Yes

Reviewer #1: The manuscript seeks to analyze Chinese influence in 49 African countries during 2018-22 through a combination of regret theory and decision-making approaches using economic indicators. The motivation is clear and commendable. But the application of regret theory or the choice of MCDM method lacks strong theoretical support. These models are typically used in behavioral decision-making contexts and there is no clear justification in the study why these methods are appropriate for testing the main hypothesis. The method seems forced and lacks empirical validation.

The paper presents an ambitious attempt to quantify Chinese influence in Africa, but it suffers from significant conceptual, methodological, and empirical weaknesses. The application of regret theory is not convincingly justified, the data is incomplete and unverified, and the chosen methodology lacks robustness in assessing geopolitical influence. The study does not significantly advance existing research on China-Africa relations, and its results are unlikely to provide useful insights for policymakers.

The key motivation of the paper suggests that with improvements in the technique, data and further refinement and redefinition of "influence" the paper could have potential.

Reviewer #2: Thank you for the opportunity to review the manuscript by Wang et al. The authors use a novel framework to capture the breadth of African nations’ responses to Chinese influence. Another major strength of the paper is in studying from the perspective of responding nations. The study holds great promise, however, the structure of the manuscript makes it difficult to follow the authors’ arguments and evaluate their findings.

Major comments:

Lack of PLOS required formatting (page numbers, line numbers) made navigating and providing feedback on this manuscript difficult. Page numbers below refer to the reviewer copy.

The structure is in line with a monograph dissertation, rather than an article. While the authors should be commended on the thorough history and literature review of the methods, that component is much more in valuable for a scoping review article (which the authors should consider splitting off) than it is a for a hypothesis-driven article. Moreover, the studies that are extensively explained in the literature review do not make an appearance in the discussion section of the manuscript, so they do not aid in the author’s interpretation.

The structure of the manuscript makes following the authors’ argument very difficult. Papers should be so logical that a reader should be able to guess your next steps. Unfortunately, the manuscript bounces between ideas that are not relevant to the study at hand. The methods section should be reordered (see comments below) to make clear what the authors are attempting. The manuscript also needs to have separate and delineated results and discussion sections.

The methods also need to be

The part of the paper that more or less lines up with a discussion section is underinterpreted, and needs to do more to help the reader understand the connection between the findings and the suggested policy implications.

Page 8: subsections in the introduction/lit review would be helpful to orient the reader.

Page 9 top: what does “without political conditions” mean? When we’re talking about the political economy of nation-state engagement strategies, any agreement’s conditions (including “purely economic” ones) could be interpreted as political conditions.

Page 9 top: what kinds of concerns?

Page 9 mid: unclear “nuanced aspects beyond conventional...”

Page 9 mid: needs to rephrase “approach aims...”. You’re describing requirements for the approach in the paragraph (since you don’t describe the approach until the next paragraph), so language should reflect that.

Overall, the combination of these 4 theories is an interesting development and it makes sense to solve specific issues that you’re trying to model. However, currently, this is lost in the structure of the paragraphs.

Page 9 bottom: MCDM, LOPCOW, MAIRCA, and Regret Theory come out of nowhere (you don’t explain until the next page). Need to motivate (with a subsection heading or one sentence) why you’re suddenly introducing this.

10: a figure illustrating how these theories are combined in a framework or a table would be really helpful to the reader. A table would allow you to more clearly compare and contrast what each theory adds to your framework, and would allow you to significantly reduce the amount of text in this section.

10: who is the actor that can have the anticipatory emotion of regret here? A nation-state is made up of people or be helmed by a person/a group of people, but does not inherently have emotions. Or unless you are describing a behavior (without ascribing emotion), which then should be clarified.

Page 10: the three paragraphs here in the middle could be condensed to a single paragraph due to redundant writing. Novelty of approach should be moved to the discussion.

Page 11: Given Table 1, the literature review in this format seems redundant. One possible solution is to cut the models that each other author has used from the text (since it’s communicated more effectively in the table). You should also try to summarize more and guide the reader to where exactly the gap in the literature is (as you do top of page 12). The remainder of the paragraph at the bottom of page 11 that starts “In 2023, several studies…” does this rather effectively.

Another solution is to cut the entire paragraph in page 11 (again because Table 1 exists) and condense the citations into the short paragraph at the top of page 12. Alternatively, you could maybe summarize the contents of page 11 in 2-5 short sentences before jumping into the paragraph on 12.

Page 12, top: this is a very effective paragraph.

Page 12, section 2.2: the entire history of and literature review of MCDM is not germane to your paper (unless you wanted to split off a scoping review paper as a second submission). The only part that is germane is the section starting with “Bounded rationality”, although this can also be dramatically condensed and summarized on how it relates specifically to MAIRCA and Regret Theory. This could then lead into the paragraph that starts near the top of page 13.

Page 13 can be condensed given Table 2.

Page 14: your paper would greatly benefit from an objective statement since it’s not clear at the top of page 14 what exactly is it that you’re trying to study (and thus difficult for the reader to evaluate the merits of the methodology).

Page 14 middle: you need to include a short sentence or paragraph explaining the general method you will employ.

Page 14 middle: would be useful in that general method paragraph to mention that you are conducting a gap analysis (this doesn’t really become clear despite a brief mention in figure 3 until page 17).

Page 14 middle: why is the explanation of regret theory literature in the methods section and not the literature review section like for the other methods? Also, this can be condensed as the reader could always go read the former paper. Section 3.1 should be cut, and section 3.2 (which I’m assuming is the authors’ novel contribution) should be the part to explain the regret theory method.

Page 17 top: Excluded countries need to be listed (fine if it’s in an appendix).

Page 17: Some background on the indicators should have been a subsection of your introduction. The reader has merely the authors’ assertion that these are vital indicators. They seem on the face like reasonable indicators, but that needs to be established in the background.

Page 19 bottom: potential reasons for FDI stock variability should have been established in the background, or, if it is a finding that is novel to your study, the potential reasons should be discussed in the discussion, not the results.

In general pages 19-22: discussion points and should not be in results, but in discussion.

Page 17-19: indicators and methods should be in the methods section, not the results section.

Figure 8: Unclear what figure 8 displays.

Minor comments:

Overall, the manuscript is too wordy. There are sentence simplifications and some condensing of paragraphs that could easily happen that would improve overall the ability of the reader to follow your argument.

Page 9 mid: Needs either “the Chinese” or “Chinese investments”

Page 20: typo in Figure 4: “standard deviation”

**Do you want your identity to be public for this peer review?** For information about this choice, including consent withdrawal, please see our Privacy Policy

Reviewer #1: No

Reviewer #2: No

---

## [Author Response · Author response to Decision Letter 1]

11 May 2025

Please see the submitted respond letter.

---

## [Decision Letter · Decision Letter 1]

9 Jul 2025

Dear Dr. Nhieu,

Thank you for submitting your manuscript to PLOS ONE. After careful consideration, we feel that it has merit but does not fully meet PLOS ONE’s publication criteria as it currently stands. Therefore, we invite you to submit a revised version of the manuscript that addresses the points raised during the review process.

We look forward to receiving your revised manuscript.

Kind regards,

Arindam Garai, Ph.D.

Academic Editor

PLOS ONE

Journal Requirements:

***Comments from the editorial office** : Upon internal evaluation of the reviews provided, we kindly request you to disregard the reviewer report provided by Reviewer 3. No amendments are required in response to reviewer 3’s comments.*

Additional Editor Comments:

I commend you on the substantial improvements made to your manuscript, particularly in terms of structure, clarity, and overall readability. Both reviewers have acknowledged these enhancements. However, several substantive issues remain that need to be addressed before the manuscript can be considered for publication. Therefore, I am recommending a major revision at this stage.

Reviewer 1 has noted that the manuscript still lacks a well-developed limitations subsection. The current paragraph at the end of the conclusion should be moved to the discussion section and significantly expanded. Additionally, a number of background statements are presented without citations, please ensure that all such claims are properly supported by references. Reviewer 1 also offers helpful technical suggestions, including improvements to figures and tables (e.g., clarifying legends and removing redundant content from Table 2).

Reviewer 2 highlights that while the topic is timely and relevant, the Introduction needs to more clearly situate your study within the broader literature and show its significance. The Methods section would benefit from further elaboration, ideally with a visual diagram outlining the research process and identifying both its strengths and limitations. The Discussion should more thoroughly interpret your results in light of prior work, drawing explicit connections to the literature. Also, the Conclusions should be revised to better reflect the policy and practical implications of your findings and to propose meaningful directions for future research, including possible links to sustainable finance.

Your manuscript demonstrates strong potential, but this is essential that these revisions, particularly those concerning citation accuracy, methodological clarity, and deeper engagement with existing research, are addressed comprehensively. I look forward to receiving your revised submission.

Reviewers' comments:

Reviewer's Responses to Questions

**Comments to the Author**

Reviewer #2: (No Response)

Reviewer #3: (No Response)

2. Is the manuscript technically sound, and do the data support the conclusions?

Reviewer #2: Yes

Reviewer #3: (No Response)

3. Has the statistical analysis been performed appropriately and rigorously?

Reviewer #2: I Don't Know

Reviewer #3: (No Response)

4. Have the authors made all data underlying the findings in their manuscript fully available?

Reviewer #2: Yes

Reviewer #3: (No Response)

5. Is the manuscript presented in an intelligible fashion and written in standard English?

Reviewer #2: Yes

Reviewer #3: (No Response)

Reviewer #2: The authors are to be commended: the manuscript is greatly improved in terms of logical flow, readability, and interpretability. However, there are still a few outstanding issues before the manuscript is ready for publication.

Major comments:

The manuscript is missing a limitations subsection in the discussion (the single paragraph at the end of the conclusion should be moved and expanded).

All statements that do not directly arise from the authors’ own analyses must be cited. This includes several particularly notable sentences listed in the line-by-line comments below.

Line-by-line comments:

1. 62: needs a citation

2. 67: needs a citation

3. 70: needs a citation

4. 100-107: these would benefit from citations showing they’ve been used or thought of in this way in the past.

5. 111: are there studies that have relied only on trade/investment as a proxy for influence? Good to cite a few here (or better yet cite a study that explains this as a limitation of previous studies)

6. Table 2: by my read, the role of table 2 is to show current state of literature which helps justify your study. The last row with your study is unnecessary for that purpose and should be removed since you already address it well in 154-156.

7. Tables 6-9: Inclusion of legends would be helpful mostly to orient the reader towards what the colors and shading mean

8. Figure 7: Repeated colors (and similar colors) make it difficult to tell which countries are which. Consider putting the 3-letter country code as a column on the left side next to each country’s starting position.

9. 411, 413: refer to the citations: which earlier findings?

10. 427-428: needs a citation

11. 430-432: is there anything you can do to corroborate this? At minimum a citation.

12. 435: needs a citation

13. 436: needs a citation

14. 437-438: needs a citation

Minor comments:

15. 100: expand since 1st appearance of acronym FDI in the text (though it shows up earlier in a figure)

Reviewer #3: This study aims to quantitatively and psychologically evaluate the varying degrees of Chinese influence across 49 African countries from 2018 to 2022 using a regret theory-based decision-making framework, thereby providing insights into the strategic economic relationships and guiding policymakers in optimizing engagement strategies. The topic presented in this work is really interesting and the autghors have done a goid work in responding the reviewer comments.hoverwr, few issue remaims:

Introduction: This section should briefly place the study in a wide context and emphasize why it is relevant carrying out the analysis. It should define the purpose of the work and its significance. In this perspective, this section is too succinct and fails to effectively point out the relevance of your contribution towards the existing literature. Some literature to consider:

https://doi.org/10.1016/j.inteco.2025.100592

https://doi.org/10.1108/JES-05-2023-0264

https://doi.org/10.1002/bse.4104

https://doi.org/10.1016/j.cogsc.2019.08.002

Materials and methods: I found this section very important for the readability of the paper. The research methodology seems underdeveloped. Methods should be described in detail. I think the research procedure could be much more clearly described by means of a diagram also highlighting its potential and limit.

Discussions: The discussion of the results is merely descriptive and the obtained evidence is flimsy due to the fact the outcomes are not supported by an adequate discussion in light of scientific literature. Authors should discuss the results and how they can be interpreted in perspective of previous studies and their implications should be discussed in the broadest context possible.

Conclusions: Conclusions must also be revised according to the previous comments. In particular, they should discuss practical and policy implications as well as future lines of research. As it stands now, they fail to extract all the juice of your work. For example is there room for sustainable finance implicatuons (https://doi.org/10.4324/9781003284703)?

I hope these comments might help in improving the paper and encourage the authors to move forward.

**Do you want your identity to be public for this peer review?** For information about this choice, including consent withdrawal, please see our Privacy Policy

Reviewer #2: No

Reviewer #3: No

---

## [Author Response · Author response to Decision Letter 2]

11 Jul 2025

Thanks for your valuable comments.

Please see our attached response letter.

---

## [Decision Letter · Decision Letter 2]

17 Oct 2025

Dear Dr. Nhieu,

Thank you for submitting your manuscript to PLOS ONE. After careful consideration, we feel that it has merit but does not fully meet PLOS ONE’s publication criteria as it currently stands. Therefore, we invite you to submit a revised version of the manuscript that addresses the points raised during the review process.

We look forward to receiving your revised manuscript.

Kind regards,

Fabien MUHIRWA

Academic Editor

PLOS ONE

Journal Requirements:

Additional Editor Comments:

Dear Authors,

Thank you for the improvements made to your manuscript. We encourage you to carefully consider the serious and interesting comments raised by reviewer 5.

Reviewers' comments:

Reviewer's Responses to Questions

**Comments to the Author**

Reviewer #2: All comments have been addressed

Reviewer #4: All comments have been addressed

Reviewer #5: All comments have been addressed

Reviewer #6: (No Response)

2. Is the manuscript technically sound, and do the data support the conclusions?

Reviewer #2: Yes

Reviewer #4: Yes

Reviewer #5: Partly

Reviewer #6: Yes

3. Has the statistical analysis been performed appropriately and rigorously?

Reviewer #2: Yes

Reviewer #4: I Don't Know

Reviewer #5: No

Reviewer #6: Yes

4. Have the authors made all data underlying the findings in their manuscript fully available?

Reviewer #2: Yes

Reviewer #4: Yes

Reviewer #5: Yes

Reviewer #6: Yes

5. Is the manuscript presented in an intelligible fashion and written in standard English?

Reviewer #2: Yes

Reviewer #4: Yes

Reviewer #5: Yes

Reviewer #6: Yes

Reviewer #2: (No Response)

Reviewer #4: Dear author,

Necessary revisions have been performed.

Manuscript can be accepted for publishing.

Best wishes.

Reviewer #5: Conceptual Framing of Influence

Although the paper explicitly treats “influence” as a multidimensional construct, the theoretical framing would benefit from greater clarity through stronger connections to contemporary debates in international political economy. In addition to economic embeddedness, the dimensions of institutional trust and asymmetrical interdependencies should be incorporated, as these factors are instrumental in explaining the persistence of Chinese influence.

Comparative Reference to Administrative Structures

The study could be further enriched by situating Chinese influence within comparative state structures. For instance, research on hierarchical governance demonstrates its role in shaping both intercity and international linkages. Incorporating evidence from studies such as “The Innovation Effect of Administrative Hierarchy on Intercity Connection” would strengthen the analysis by showing how administrative arrangements mediate influence dynamics.

Trust and Decision-Making Models

Given the framework’s integration of regret theory and bounded rationality, the link between trust (or mistrust) among actors and their perceptions of regret and opportunity costs should be made more explicit. Reference to studies such as Jiang, Zhao, Dong, & Hu (2024) doi: https://doi.org/10.1016/j.inffus.2023.102173 would enhance the behavioral dimension of the model by embedding trust within decision-making frameworks.

Insufficient Discussion of Labor Regulations

While addressing Chinese labor flows into Africa, the manuscript overlooks the regulatory and governance frameworks shaping such dynamics. A deeper analysis of how labor law regimes condition socio-economic outcomes would add value. Incorporating works such as “The Effect of Changing Criminal Laws on the Regulation of Labor Conflicts in P.R. China” would provide the necessary legal context.

The Graduate Workforce and Human Capital Formation

The analysis does not sufficiently engage with the role of African human capital in shaping reciprocal relations with China. Expanding the discussion to consider labor mobility, graduate employment, and structural dependencies would be beneficial. For instance, Zhao, Chang, & Yusof (2023) offer useful insights in “Analysis of Job Search Success Factors among Chinese University Graduates A Pathway towards Employment-oriented Linkages.”

Supply Chain Risk Dimension

By focusing primarily on FDI, loans, and infrastructure, the paper underplays global supply chain vulnerabilities and their implications for African economies. Including a resilience-oriented perspective such as risks of supply disruptions and financial contagion would provide a more comprehensive assessment. Relevant methodological insights can be drawn ) on risk-based MCDM approaches, :Risk Identification and Prioritization in China's New Energy Vehicle Supply Chain: An Integrated Tanimoto Similarity and Fuzzy-DEMATEL Approach).

Methodological Transparency

The integration of regret theory is innovative; however, the choice of parameters (e.g., α = 0.88, β = 0.3) lacks sufficient justification. The authors should clarify whether these were derived from prior literature, expert elicitation, or sensitivity analysis. Testing robustness under alternative parameter values would further strengthen confidence in the findings.

Validation of Results

To ensure robustness and mitigate method-specific bias, the presented rankings should be validated using additional MCDM techniques such as TOPSIS, VIKOR, or fuzzy-DEMATEL, refer to doi: https://doi.org/10.1016/j.ijme.2024.101018.; doi: 10.15837/ijccc.2022.6.5010., doi: https://doi.org/10.1016/j.solener.2024.112692.

This triangulation would demonstrate that the results are not overly sensitive to the selected model.

Integration of Qualitative Insights

The study is heavily structural and primarily quantitative, with limited incorporation of qualitative evidence. Triangulating the quantitative findings with case studies from contexts such as Kenya, Nigeria, or Ethiopia would enhance interpretability and increase policy relevance.

Policy Implications

The policy recommendations are currently broad. It would be valuable to distinguish between immediate operational options (loan conditions, export market diversification) and long-term strategic imperatives (institutional capacity-building, governance reform). This would yield more actionable guidance for policymakers.

Clarity of Presentation

Several aspects of presentation require improvement:

(a) equations should be linked to illustrative examples and explained intuitively,

(b) figures and tables should include more detailed captions with explicit reference to policy relevance, and

(c) the discussion should more clearly integrate theoretical contributions with empirical findings to delineate the study’s novelty.

Reviewer #6: For the Manuscript Number (PONE-D-25-05084R2), Title “ Chinese Influence in Africa by Integrated Regret Theory and Multi-criteria Decision Analysis”, I suggest to be accepted based on the following points:

1- Authors have revised and updated the introduction and literature properly.

2- The effort is clearly beneficial to the community.

3- The manuscript has also precision and creative thinking.

4- The quality of work that was put into researching and presenting the methodology is commendable.

5- There have been a number of improvements made to the results analysis's level of detail.

**Do you want your identity to be public for this peer review?** For information about this choice, including consent withdrawal, please see our Privacy Policy

Reviewer #2: No

Reviewer #4: No

Reviewer #5: No

Reviewer #6: **Yes: ** Husam Jasim Mohammed

---

## [Author Response · Author response to Decision Letter 3]

20 Oct 2025

Thank for your valuable comments.

However, our study may have several diference with your point-of-view.

Please see our point-to-point response letter.

---

## [Editor Report · Decision Letter 3]

29 Oct 2025

Chinese Influence in Africa by Integrated Regret Theory and Multi-criteria Decision Analysis

PONE-D-25-05084R3

Dear Dr. Nhieu,

We’re pleased to inform you that your manuscript has been judged scientifically suitable for publication and will be formally accepted for publication once it meets all outstanding technical requirements.

Kind regards,

Fabien MUHIRWA

Academic Editor

PLOS ONE
---

## [Editor Report · Acceptance letter]

PONE-D-25-05084R3

PLOS ONE

Dear Dr. Nhieu,

I'm pleased to inform you that your manuscript has been deemed suitable for publication in PLOS ONE. Congratulations! Your manuscript is now being handed over to our production team.

Kind regards,

on behalf of

Dr. Fabien MUHIRWA

Academic Editor

PLOS ONE